# Pooled clone collections by multiplexed CRISPR-Cas12a-assisted gene tagging in yeast

Benjamin C. Buchmuller [1,4], Konrad Herbst [1,4], Matthias Meurer[1], Daniel Kirrmaier[1,2], Ehud Sass[3], Emmanuel D. Levy [3] & Michael Knop [1,2]

Clone collections of modified strains ("libraries") are a major resource for systematic studies with the yeast *Saccharomyces cerevisiae*. Construction of such libraries is time-consuming, costly and confined to the genetic background of a specific yeast strain. To overcome these limitations, we present CRISPR-Cas12a (Cpf1)-assisted tag library engineering (CASTLING) for multiplexed strain construction. CASTLING uses microarray-synthesized oligonucleotide pools and in vitro recombineering to program the genomic insertion of long DNA constructs via homologous recombination. One simple transformation yields pooled libraries with >90% of correctly tagged clones. Up to several hundred genes can be tagged in a single step and, on a genomic scale, approximately half of all genes are tagged with only ~10-fold oversampling. We report several parameters that affect tagging success and provide a quantitative targeted next-generation sequencing method to analyze such pooled collections. Thus, CASTLING unlocks avenues for increasing throughput in functional genomics and cell biology research.

---

[1] Zentrum für Molekulare Biologie der Universität Heidelberg (ZMBH), DKFZ-ZMBH Alliance, 69120 Heidelberg, Germany. [2] Cell Morphogenesis and Signal Transduction, German Cancer Research Center (DKFZ), DKFZ-ZMBH Alliance, 69120 Heidelberg, Germany. [3] Department of Structural Biology, Weizmann Institute of Science, Rehovot 7610001, Israel. [4] These authors contributed equally: Benjamin C. Buchmuller, Konrad Herbst. Correspondence and requests for materials should be addressed to M.K. (email: m.knop@zmbh.uni-heidelberg.de)

The systematic screening of arrayed biological resources in high-throughput has proven highly informative and valuable to disentangle gene and protein function. For eukaryotic cells, a large body of such data has been obtained from yeast strain collections ("libraries") in which thousands of open-reading frames (ORFs) are systematically altered in identical ways, for example, by gene inactivation or over-expression to determine gene dosage phenotypes and genetic interactions[1–3]. Likewise, gene tagging, for example with fluorescent protein reporters, has been used in functional genomics to study protein abundance[4], localization[5], turnover[6,7], or protein–protein interactions[8–10].

Due to their genewise construction, producing arrayed clone collections is typically time-consuming and cost-intensive. For yeast, this has been partly addressed with the development of SWAT libraries in which a generic N- or C-terminal tag can be systematically replaced with the desired reporter for tagging any ORF in the genome[11,12]. However, manipulation and screening of arrayed libraries remains dependent on special equipment to handle the strain collections and is confined to the genetic background of the yeast strain BY4741[13] in which most of these libraries were constructed. Therefore, arrayed libraries cannot address current and future demands in functional genomics that embrace the systematic analysis of complex traits or the comparison of different strains or species[14].

We imagine that a paradigm shift from arrayed to pooled library generation may offer a solution: experimentation with pooled biological resources is already well established[15] and the phenotype-to-genotype relationship can be inferred conveniently by genotyping phenotypically distinct subsets of pooled libraries using next-generation sequencing (NGS).

To generate the pooled libraries rapidly and independent of their genetic background, an efficient strategy to introduce the genetic alterations is required. For example, RNA-programmable CRISPR (clustered regularly interspaced short palindromic repeat)-associated endonucleases have revolutionized the creation of pooled collections of gene activation and inactivation mutants in mammalian cells[16–18] since thousands of CRISPR guide RNAs (gRNAs) can be produced by cost-effective microarray-based oligonucleotide synthesis. In bacteria and yeast, strategies that exploit homologous recombination have enabled multiplexed gene editing by delivering short repair templates on the same oligonucleotides as the gRNA[19–22] with applications for phenotypic profiling of genomic sequence variations.

Because of high-throughput, low-cost, and broad host versatility, it is interesting to leverage these CRISPR-based methods beyond loss- or gain-of-function screens for the precise insertion of longer DNA constructs that deliver reporter molecules or tags to monitor the different cellular components encoded in the genome. Rapid access to such collections would synergize, for example, with image-activated cell sorting[23], and enable to use subcellular localization as a criterion for cell sorting.

To exert gene tagging in a pooled format, thousands of DNA constructs must be generated, each containing the reporter gene flanked with locus-specific homology arms and paired with a corresponding gRNA. However, parallel construction of thousands of such constructs is challenging.

Here, we describe "CRISPR-assisted tag library engineering" (CASTLING) to create pooled collections of hundreds to thousands of yeast clones in a single reaction tube. All clones contain the same, large DNA construct (up to several kb in length) accurately inserted at a different, yet precisely specified chromosomal locus. CASTLING is compatible with microarray-based oligonucleotide synthesis since each insertion is specified by a single oligonucleotide only. Our method employs an intramolecular recombineering procedure that allows the conversion of oligonucleotide pools into pools of tagging cassettes.

In this proof-of-concept study, we establish CASTLING in the yeast *Saccharomyces cerevisiae* using gene tagging with fluorescent protein reporters as an example. We derive a set of rules to aid designing effective CRISPR RNAs (crRNAs) for the CRISPR endonuclease Cas12a (formerly known as Cpf1)[24] for C-terminal tagging of genes in yeast, and determine parameters to maximize tagging success in libraries of different sizes. We use a simple assay based on fluorescence-activated cell sorting (FACS) to demonstrate how CASTLING libraries can be used for proteome profiling and ad hoc characterization of previously uncharacterized proteins, and provide a targeted NGS method for the quantitative analysis of such pooled experiments.

## Results

**Gene tagging with SICs.** The main component of CASTLING is a linear DNA construct that comprises multiple genetic elements: the "feature" for genomic integration such as a fluorescent protein tag, a selection marker, a gene for a locus-specific Cas12a crRNA and flanking homology arms to direct the genomic insertion of the DNA fragment by homologous recombination (Fig. 1a). We conceptually termed these DNA constructs "self-integrating cassettes" (SICs).

We used Cas12a from *Francisella novicida U112* (FnCas12a), which is functional in yeast[25], because the genomic target space of the Cas12a endonucleases is defined by A/T-rich protospacer-adjacent motifs (PAMs)[26–29]. This makes Cas12a endonucleases well suited for genetic engineering at transcriptional START and STOP sites in many organisms (Supplementary Fig. 1).

To test the SIC strategy, we generated SICs for tagging several highly expressed genes with a fluorescent protein reporter. After individual transformation of the SICs and marker selection, we obtained 100–1000 times more colonies from hosts that had transiently expressed a Cas12a endonuclease as compared to a host that did not (Fig. 1b). Also, the presence of a crRNA gene specific for the target locus of the SIC was required (Supplementary Fig. 2), indicating that a functional crRNA transcribed from the linear DNA fragment promotes the integration of a SIC. Based on fluorescent colony counts, tagging fidelity had increased from 50 to 85% in the absence of Cas12a to 95–98% when recombination was stimulated by the action of Cas12a (Fig. 1b).

We also tested Cas12a endonucleases from other species[24], finding that Cas12a from *Acidaminococcus* sp. BV3L6 (AsCas12a) showed similar activity as FnCas12a (Supplementary Fig. 3a–c). However, we continued with FnCas12a since it offered a broader genomic target space in the yeast genome than AsCas12a (Supplementary Fig. 4).

Because of the high efficiency of SIC integration, we worried that multiple loci could be tagged within the same cell when different SICs were transformed as pools. We therefore transformed a mixture of two SICs, one to tag *ENO2* with mCherry and the other one to tag *PDC1* with sfGFP. We detected only a few individual colonies where both genes were fluorescently tagged (Fig. 1c), independent of the relative concentration of the two SICs used for transformation (Fig. 1d). Therefore, tagging multiple loci in the same cell would rarely occur if more than one SIC was transformed simultaneously.

**Implementing CASTLING for pooled gene tagging.** To produce many different SICs in a pooled format using microarray-synthesized oligonucleotides, all gene-specific elements of a SIC, that is, the crRNA sequence and both homology arms, must be contained in a single oligonucleotide—one for each target locus (Fig. 2a). In turn, this demands a strategy to convert these oligonucleotides in bulk into the corresponding SICs.

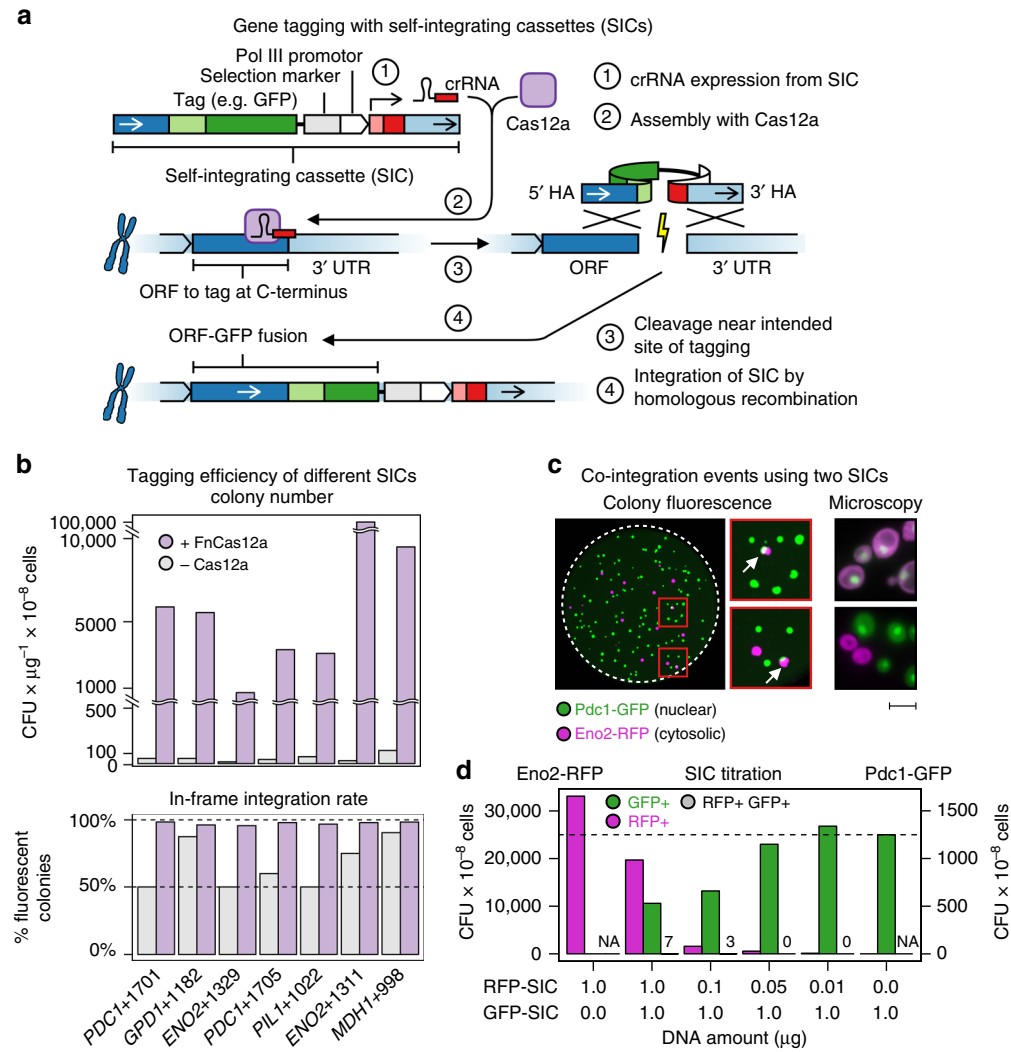

**Fig. 1** CRISPR-Cas12a-assisted single gene-tagging in yeast. **a** After transformation of the self-integrating cassette (SIC) into a cell, the CRISPR RNAs (crRNA) expressed from the SIC directs a CRISPR-Cas12a endonuclease to the genomic target locus where the DNA double strand is cleaved. The lesion is repaired by homologous recombination using the SIC as repair template so that an in-frame gene fusion is observed. **b** Efficiency of seven SICs of C-terminal tagging of highly expressed open-reading frames (ORFs) with a fluorescent protein reporter, in the absence (gray) or presence (purple) of *Francisella novicida U112* (FnCas12a). Colony-forming units (CFUs) per microgram of DNA and cells used for transformation, and integration fidelity by colony fluorescence are shown. **c** Co-integration events upon simultaneous transformation of two SICs directed against either *ENO2* or *PDC1*. Both SICs confer resistance to Geneticin (G-418), but contain different fluorescent protein tags. Colonies exhibiting green and red fluorescence (arrows) were streaked to identify true co-integrands. False-color fluorescence microscopy images show nuclear Pdc1-GFP (green fluorescent protein) in green and the cytosolic Eno2-RFP in magenta; scale bar 5 μm. **d** Titration of both SICs against each other (lower panel) with evaluation of GFP-tagged (GFP+), red fluorescent protein (RFP)-tagged (RFP+) or co-transformed (GFP+ RFP+) colonies. **b**–**d** Source data are provided as a Source Data file

We implemented a three-step molecular recombineering procedure for this conversion that is executed in vitro (Fig. 2b, Supplementary Fig. 5a–e). Its central intermediate is a circular DNA species formed by the oligonucleotides and a feature cassette. The feature cassette provides all the generic elements of the SIC, that is, the tag (e.g. green fluorescent protein (GFP)), the selection marker and an RNA polymerase III (Pol III) promoter to express the crRNA. The circular intermediates are then amplified by rolling circle amplification (RCA) instead of PCR to avoid the formation of chimeras containing non-matching homology arms. The individual SICs are finally released by cleaving the DNA concatemer using a restriction site in between both homology arms.

To accommodate all gene-specific elements on a single oligonucleotide, it was critical to use a Cas12a endonuclease because its crRNA consists of a comparably short direct repeat sequence (~20 nt) that precedes each target-specific CRISPR spacer (~20 nt; Supplementary Fig. 5f). This arrangement allows the Pol III promoter, which drives crRNA expression, to remain part of the feature cassette, while the short Pol III terminator[30] can be included in the oligonucleotide itself. This design leaves enough space for homology arms of sufficient length for homologous recombination (>28 bp)[31]. Adding up all the sequences, each oligonucleotide (160–170 nt) is within the length limits for commercial microarray-based synthesis.

To select CRISPR targets near the desired chromosomal insertion points and to assist the design of the oligonucleotide sequences for microarray synthesis (Supplementary Fig. 6a–d), we wrote the software tool castR (https://github.com/knoplab/castR/tree/v1.0). For use with small genomes, castR is available online (http://schapb.zmbh.uni-heidelberg.de/castR/).

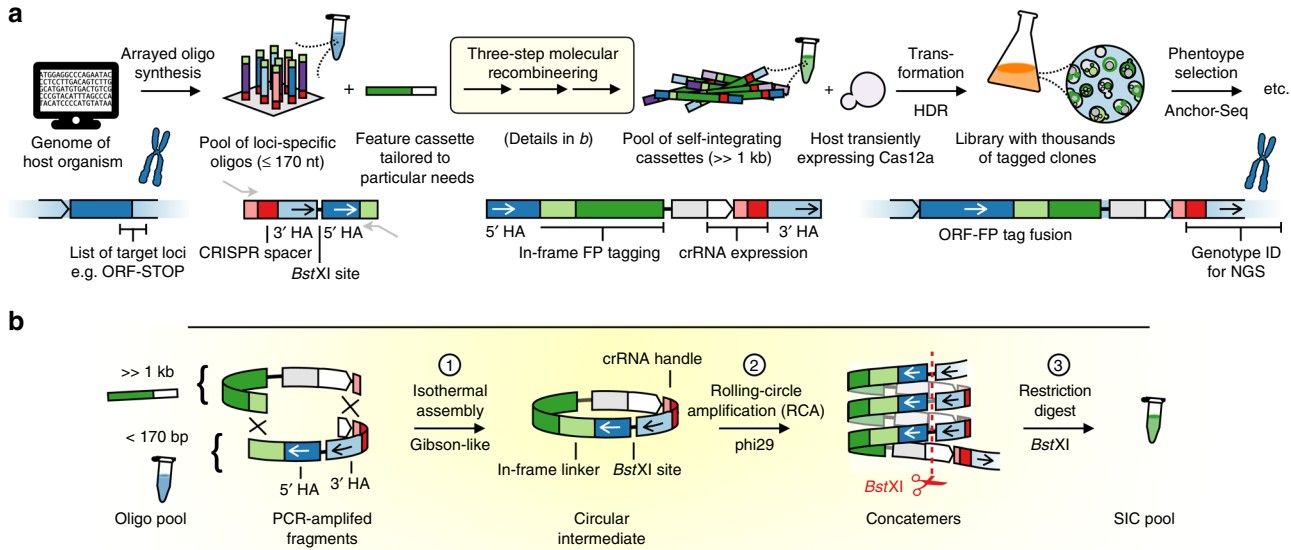

**Fig. 2** CRISPR-Cas12a (Cpf1)-assisted tag library engineering (CASTLING) in a nutshell. **a** For each target locus, a DNA oligonucleotide with site-specific homology arms (HAs) and a CRISPR spacer encoding a target-specific CRISPR RNAs (crRNA) is designed and synthesized as part of an oligonucleotide array. The resulting oligonucleotide pool is recombineered with a custom-tailored feature cassette into a pool of self-integrating cassettes (SICs). This results in a clone collection (library) that can be subjected to phenotypic screening and genotyping, for example, using Anchor-Seq[12]. **b** The three-step recombineering procedure for SIC pool generation; details are given in the main text and Methods

**Using CASTLING to generate a GFP library of nuclear proteins.** To test CASTLING, we sought to create a small library covering a set of proteins with known localization[32]. We chose 215 nuclear proteins whose localization had been validated in different genome-wide data sets[12,33]. We designed 1577 oligonucleotides covering all suitable PAM sites within 30 bp around the C-termini of the selected ORFs, yielding seven oligonucleotides per gene on average. We purchased this oligonucleotide pool three times from different suppliers, one pool from supplier A (pool A) and two pools from supplier B (pools B1 and B2; Fig. 3a, Methods). The amount of starting material for PCR to amplify each pool was adjusted to obtain a product within ~20 cycles. We observed that pool A required about 200-fold more starting material than pool B1 or B2 (Fig. 3a). After recombineering with a feature cassette comprising the bright green fluorescent protein reporter mNeonGreen[34], we generated four different libraries in technical duplicates of 30,000–95,000 clones each (Fig. 3a, Supplementary Table 1).

We used NGS in combination with unique molecular identifiers (UMIs)[35] to quantitatively analyze the entire procedure at three stages: after PCR amplification of the oligonucleotide pool, after SIC amplification (Supplementary Fig. 7a), and after yeast library construction. To characterize the yeast libraries, we adapted the targeted NGS method Anchor-Seq[12] with UMIs to analyze the CRISPR spacers of the inserted SICs along with the genomic sequence adjacent to the insertion site in all clones of the libraries (Supplementary Fig. 7b).

Overall, the represented oligonucleotide diversity gradually decreased during recombineering (Fig. 3b). The best performance was observed in one duplicate generated from pool B2 that used a high amount of starting material (libraries 4a and 4b), preserving more than 70% of the originally amplified oligonucleotides in the SIC pool and more than 60% of the oligonucleotide diversity in the yeast libraries (Fig. 3b). This loss in complexity was alleviated by the fact that multiple oligonucleotides were included per gene and we observed that more than 90% of the targeted genes were tagged in library 4a and 4b (Fig. 3c). We noticed that low abundant oligonucleotides after PCR amplification were prone to depletion during SIC preparation, accounting for the observed

loss in sequence diversity (Fig. 3d). Across all preparations, copy numbers of individual oligonucleotides were highly correlated between duplicates after PCR (Pearson's correlation >0.96), but less between synthesis replicates (0.78–0.90), and least for oligonucleotide pools obtained from different suppliers (Fig. 3e). After recombineering and RCA, no significant correlation of SIC copy numbers was observed except for libraries 4a and 4b. A more detailed analysis indicated that 50% of the sequences exhibited a copy number change >2-fold during RCA (Fig. 3f), which could explain the loss of correlation between replicates after RCA. Taken together, these analyses identified the quality and amount of starting material and its recovery during recombineering as critical factors to preserve library diversity. Nevertheless, for a small library of 215 genes, CASTLING enabled tagging most of the selected genes within one library preparation.

Next, we quantified tagging fidelity by fluorescence microscopy, which was possible because we had selected genes encoding proteins with validated nuclear localization: 90–95% of the cells had a nuclear localized mNeonGreen signal in all libraries (Fig. 3g, h). The remainder of the cells showed either no fluorescence (2–8%) or a fluorescence signal elsewhere (0–4%), usually in the cytoplasm with one exception (see below). So, nearly all genes must have been tagged in the correct reading frame.

For the clones with no fluorescence signal, we suspected either frameshift mutations in the polypeptide linker (due to faulty oligonucleotides) or in the fluorescent protein reporter (due to limited fidelity of DNA polymerases), or off-target integration of the SIC. Sequencing of several insertion junctions of dark clones revealed small deletions of one or more nucleotides in the 5′-homology arms that direct the SICs to the 3′ ends of the ORFs. Therefore, the majority of dark clones appeared to contain correctly targeted SICs in which mNeonGreen was not in frame due to errors in the sequences derived from the oligonucleotides.

Next, we generated library-wide Anchor-Seq data encompassing the crRNA sequences and the 3′-insertion junctions. This identified 280 instances in which the crRNA sequence and the genomic insertion site did not match. These off-target insertions corresponded to <0.2% of the clones. Most of them were single

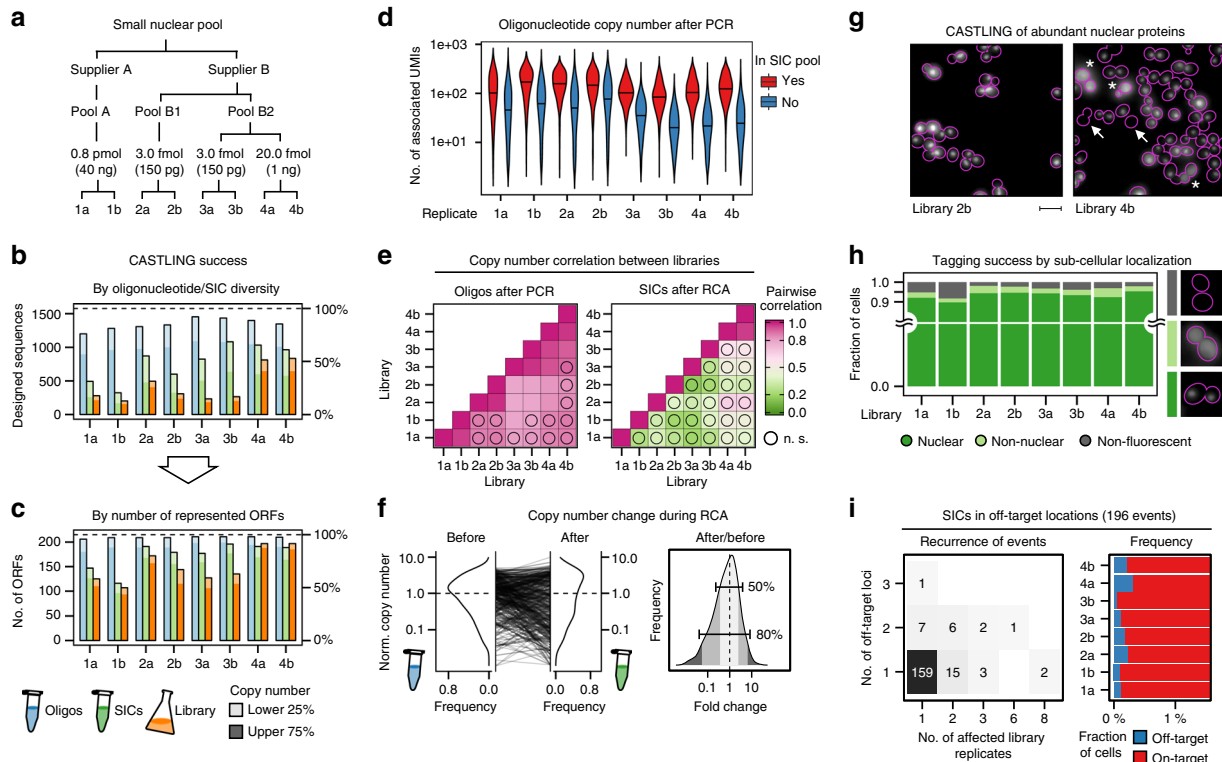

**Fig. 3** CRISPR-Cas12a (Cpf1)-assisted tag library engineering (CASTLING) for tagging 215 nuclear proteins with a green fluorescent protein. **a** Three oligonucleotide pools of the same design (1577 sequences, Supplementary Table 1) were used to create four tag libraries by CASTLING in duplicate sampling the indicated amount of starting material for PCR. **b** Detected oligonucleotide sequences of the design after PCR amplification (blue), self-integrating cassette (SIC) assembly (green), and in the final library (orange); oligonucleotides with copy number estimates (unique molecular identifier (UMI) counts) in the lowest quartile (lower 25%) are shown in light shade. **c** Same as **b**, but evaluated in terms of open-reading frames (ORFs) represented by the oligonucleotides or SICs. **d** Copy number of PCR amplicons recovered (red) or lost (blue) after recombineering; black horizontal lines indicate median UMI counts. **e** Pearson's pairwise correlation of oligonucleotide or SIC copy number between replicates after PCR or rolling circle amplification (RCA), respectively; n.s., not significant ($p > 0.05$). **f** Kernel density estimates of copy number in replicate 1a as normalized to the median copy number observed in the oligonucleotide pool (before recombineering) and after recombineering into the SIC pool (left panel); the distribution of fold changes (right panel) highlights two frequency ranges: [0.1–0.9], that is, 80% of SICs, and [0.25–0.75], that is, 50% of SICs. **g** Representative fluorescence microscopy images of cells displaying nuclear, diffuse non-nuclear (asterisks), or no mNeonGreen fluorescence (arrows); scale bar 5 μm. **h** Quantification of fluorescence localization in >1000 cells in each replicate. **i** Recurrence of off-target events as revealed by Anchor-Seq across all library replicates and all genomic loci (left panel); the fraction of cells with SICs integrated at off-target sites (blue) within each clone population (red) is shown (right panel, axis trimmed). **b**–**i** Source data are provided as a Source Data file

occurrences associated with 196 different SICs in total. Only 37 SICs showed off-target insertion at various genomic loci or in more than one library replicate (Fig. 3i). It remains, however, unclear which of these insertions were caused by Cas12a-mediated cleavage at an off-target site and which were spontaneous chromosomal insertions.

In addition to these events, we observed fluorescence signals at unexpected subcellular localizations. For example, 2% of the cells in library 2b displayed fluorescence at the spindle-pole body, which we attributed, based on Anchor-Seq, to a *TEM1*-mNeon-Green gene fusion. Indeed, on average 1.6% of all cells across all libraries had integrated SICs originally designed for another experiment in this study, which must have entered SIC or library preparation as a result of contamination.

Together, these experiments demonstrate that in a pooled experiment CASTLING allows for highly efficient tagging of hundreds of genes with low levels of off-target insertion.

**Parameters affecting tagging success on a genome-wide scale.** Simultaneously with the small pool of nuclear proteins, we designed an oligonucleotide pool for C-terminal tagging of the yeast proteome. For crRNA design, we first retrieved a set of more

than 34,000 candidate CRISPR targets using our castR script and using TTV (V = A, C, or G) and TYN (Y = C or T; N = any nucleobase) as PAMs. Next, we removed sequences that contained thymidine runs longer than five nucleotides, since they may prematurely terminate Pol III transcription[30]. Subsequently, we filtered out crRNA targets with a high off-target estimate and removed most, but not all, target sequences that are not destroyed after insertion of the SICs (Supplementary Note 1). From the remainder, we chose randomly 12,472 sequences (limited by the chosen microarray) that covered 5664 of 6681 (85%) of the annotated ORFs in *S. cerevisiae*[36]. Although the number of oligonucleotides per gene was lower as compared to the nuclear pool, the high number of genomic targets should allow identifying parameters that would influence tagging success and clone representation in such large-scale experiments.

After PCR and SIC pool generation, we sequenced the PCR amplicons and one SIC pool. We analyzed the sequencing data implementing a de-noising strategy to discriminate errors introduced during NGS from errors in the templates[37]. This revealed that the PCR product contained 57% of the designed oligonucleotides, but only 31% of the designed sequences were represented by at least one error-free amplicon. Similarly, 51% of

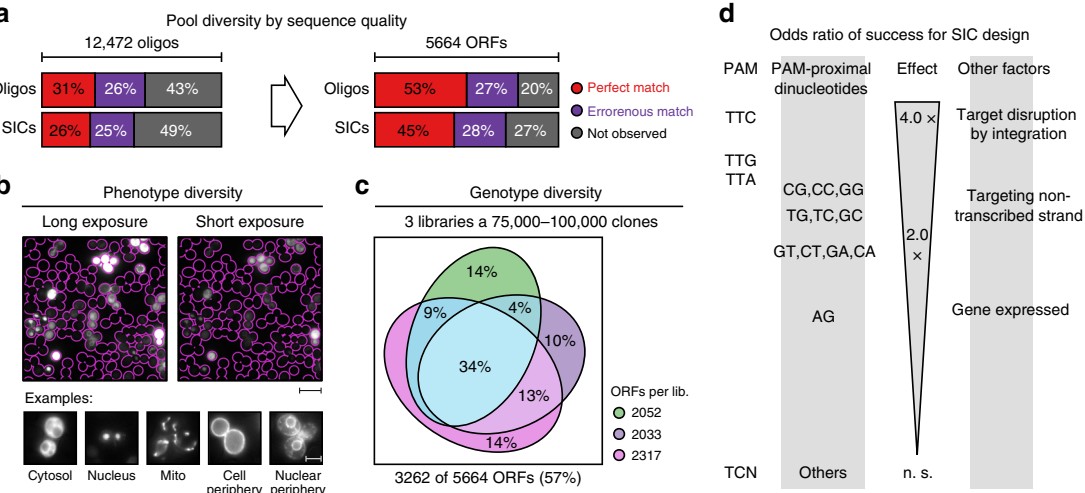

**Fig. 4** Identification of factors influencing clone representation in CRISPR-Cas12a (Cpf1)-assisted tag library engineering (CASTLING). **a** Sequence quality of an oligonucleotide pool (oligonucleotide pool C, Supplementary Table 2) after PCR amplification and self-integrating cassette (SIC) assembly. Following de-noising of next-generation sequencing (NGS) artifacts, molecules that aligned with any of the 12,472 designed oligonucleotides were classified error-free, erroneous, or absent at the respective stage (left panel). The genotype space (designed: 5664 open-reading frames (ORFs)) was covered by each class (right panel). **b** Representative fluorescence microscopy images of a pooled tag library (derived from oligonucleotide pool C); scale bar: 20 µm (overview), 5 µm (details). **c** Genotype diversity within three independent library preparations (libraries #1.1, #1.2, and #1.3, Supplementary Table 2) generated from the same oligonucleotide pool; all libraries combined tagged 3262 different ORFs. **d** Summary of parameters significantly (Fisher's exact test, $p < 0.05$) increasing the likeliness of tagging success beyond SIC abundance (details in Supplementary Fig. 7a–b). **a–c** Source data are provided as a Source Data file

all designed sequences were detected in this SIC pool, but only 25% were error free (Fig. 4a). Due to redundancy, the error-free SICs in this pool still covered 45% of the 5664 ORFs.

To explore how many genes could be tagged with this oligonucleotide pool, we repeated PCR and SIC assembly three times. Following transformation in yeast, this resulted in three independent libraries of 75,000–100,000 clones each. Inspection of the cells by fluorescence microscopy revealed localization across a broad range of subcellular compartments (Fig. 4b). By Anchor-Seq, we detected a total of 3262 different ORFs (58% of all targeted ORFs), of which 1127 ORFs (20%) were shared across all replicates (Fig. 4c, Supplementary Table 2).

The acquired data allowed us to identify factors that might have impeded efficient genomic integration of a SIC. First, the likelihood of tagging success was 3- to 4-fold decreased when the crRNA target sequence was not disrupted by the inserted SIC, that is, when recurrent cleavage of the locus was possible. Neither nucleosome occupancy of the PAM nor of the target sequence itself had a statistically significant impact on the tagging success in this library. However, the choice of the PAM (TTC > TTG > TTA » TYN) and the first two PAM-proximal nucleotides (CG, CC, GG) increased the chances of target integration 2- to 3-fold each (Fig. 4d, Supplementary Fig. 8a–b). Interestingly, it seemed advantageous to target genes on their non-transcribed strand by Cas12a. Despite the limited success to create a genome-wide library at first trial, we anticipated that these parameters could help to improve tagging success for CASTLING in yeast.

**Using CASTLING to construct complex pooled yeast libraries**. To further investigate the creation of genome-wide pooled libraries with CASTLING, we designed a new microarray for tagging 5940 ORFs. Applying these rules for each ORF, we selected 17,691 target sites near the STOP codon and filled up the remaining positions on a 27,000-well array. We generated three libraries in total using two different strategies to investigate

the minimal effort that would be required for creating a large library with CASTLING.

First, we pooled SICs from 30 RCAs and generated a large library of 704,000 clones (LibA), and a small library of 44,000 clones (LibB). Second, we constructed a third library of 116,000 clones (LibC) using a SIC pool made from two RCAs of the same oligonucleotide pool (Fig. 5a). To quantify genotype composition in each of the different libraries, we again used Anchor-Seq at the crRNA junction. Altogether, the three libraries contained tagged alleles of 76% of all the targeted ORFs with an overlap of 43% between the three libraries (Fig. 5b, c). The largest library, LibA, contained the most tagged ORFs (3801 ORFs), corresponding to 64% of the design. Interestingly, the much smaller library LibB with 44,000 clones already contained 80% of these genotypes. LibB and LibC each covered ~50% of the desired ORFs, sharing 2038 ORFs. In practical terms, this implied that about one-third of the intended genes could be reliably and reproducibly tagged with minimal effort by recovering 40,000–120,000 clones only.

We validated the rule set used for oligonucleotide design by comparing SICs with approximately equal copy number in the SIC pool (Supplementary Fig. 9a–b).

Functional studies that use pooled libraries fundamentally depend on enrichment procedures to physically separate cells based on the information provided by the reporter. When fluorescent protein fusions to endogenous proteins are used, high-resolution fluorescence microscopy would be the method of choice, as this would enable scoring and subsequent cell sorting based on very complex but highly informative phenotypes. The necessary technology is currently under development[23]. To demonstrate that CASTLING libraries can be used for screening, we reverted to FACS, which permits sorting based on fluorescence intensity.

Starting from a library containing 2052 mNeonGreen-tagged ORFs (Fig. 4c), we first sorted cells for which fluorescence could be detected by FACS. Anchor-Seq revealed that in comparison to the starting library, this cell population contained 848 genotypes, while 732 genotypes were depleted. Therefore, we estimated that

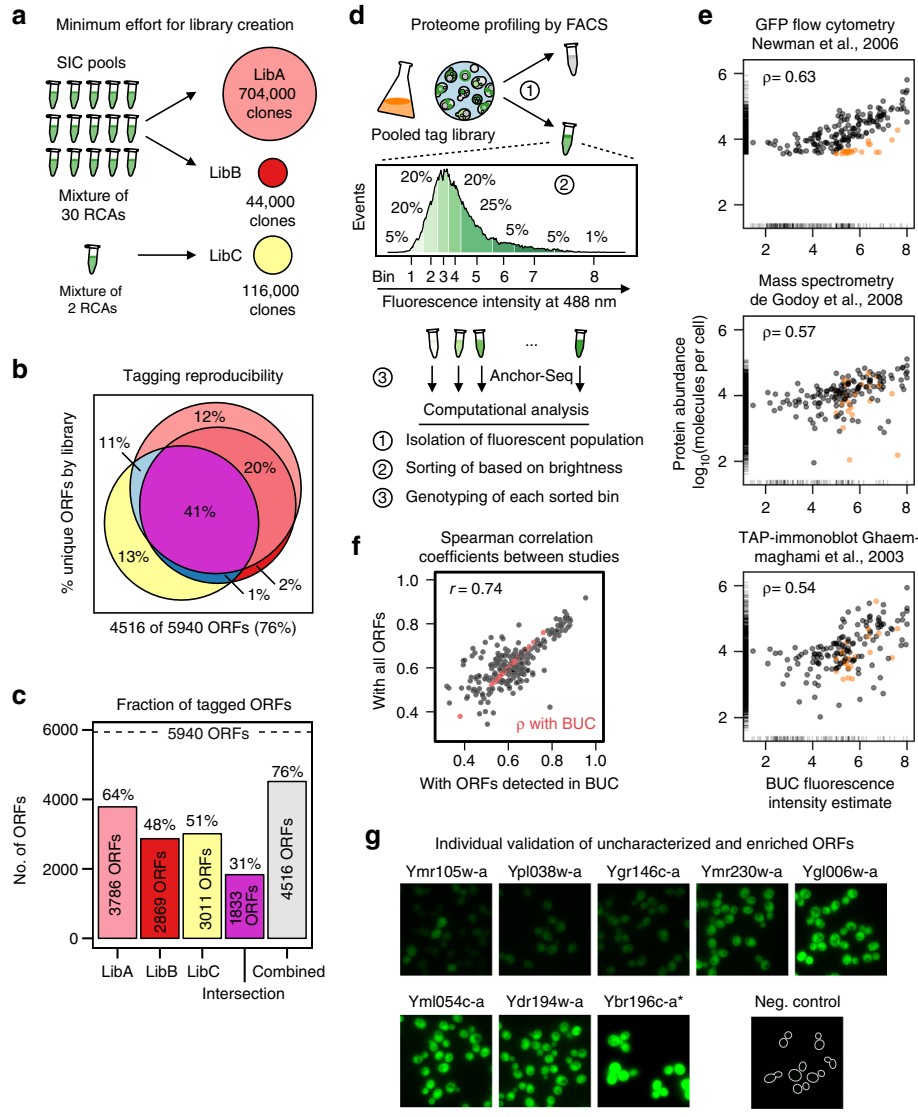

**Fig. 5** Creating and screening large CRISPR-Cas12a (Cpf1)-assisted tag library engineering (CASTLING) libraries. **a** Three libraries with different numbers of collected clones were generated from self-integrating cassette (SIC) pools combining either 2 or 30 recombineering reactions to investigate the minimum effort for a proteome-wide (design: 5940 open-reading frames (ORFs), oligonucleotide pool D, Supplementary Table 2) CASTLING library (details in Methods). **b** Venn diagram of genotypes recovered in each of the three libraries; all libraries combined tagged 4516 different ORFs. **c** Genotype diversity in each of the three libraries, shared between them, or after their combination. **d** Proteome profiling by fluorescence intensity of a non-exhaustive mNeonGreen tag library (library #1.1, Fig. 4c, Supplementary Table 2) using fluorescence-activated cell sorting (FACS). After enriching the fluorescent sub-population of the library and determining the fold enrichment of each genotype by next-generation sequencing (NGS), this sub-population was sorted into eight bins according to fluorescent intensity. Analysis of each bin by Anchor-Seq and on-site nanopore sequencing allowed the assignment of an expected protein abundance for each genotype. **e** Pairwise comparisons between fluorescence intensity estimates calculated from genotype distribution across all bins (Methods, Eq. 2; this study denoted as BUC) and protein abundances reported by selected genome-scale experiments[4,39,40] normalized to molecules per cell[38]. Outliers (orange) were determined based on the comparison to a green fluorescent protein (GFP) tag flow cytometry study[39]. Spearman's correlation coefficients ($\rho$) are given. Marginal lines indicate abundance estimates only present in the respective study but missing in the other.
**f** Comparison of Spearman's correlation coefficients between studies either considering their overlap in detected ORFs or only the overlap with the 435 ORFs we could detect in this experiment. A Pearson's correlation coefficient ($r$) is given. **g** Eight genes that had not been characterized in other genome-scale experiments[38] were tagged individually to verify whether fluorescence intensity corresponded with their predicted characterization by FACS. Same exposure time for all fluorescent microscopy images except for Ybr196c-a, which was imaged at 10% excitation; scale bar 10 µm. **b**, **c**, **e** Source data are provided as a Source Data file

35% of the mNeonGreen-tagged genes could be profiled based on fluorescence intensity in our pooled study, which agrees with a meta-analysis on yeast protein abundance[38] that reported abundance estimates for 1404 proteins characterized by flow cytometric fluorescence measurements[39], that is, 34% of 4159 ORFs tagged in the C-GFP library[32].

To determine the fluorescence intensity of individual proteins in the fluorescence-enriched fraction, we sorted the cells into eight fractions of increasing fluorescence intensity Next, we analyzed the genotype distribution within the bins using Anchor-Seq. We sequenced the amplified insertion junctions using MinION nanopore sequencing. This method allows a more rapid

profiling workflow, but provides a lower sequencing depth as compared to Illumina dye sequencing, which we usually used to characterize CASTLING libraries. We obtained 18,638 informative reads, which enabled us to determine the relative enrichment in the individual bins for 435 (50%) of the 848 tagged proteins. These estimates correlated well with the abundance estimates from the flow cytometry study by Newman et al. [39] (Spearman's correlation coefficient >0.63; Fig. 5e) and were comparable with different protein abundance data sets consolidated by Ho et al. [38] (Supplementary Fig. 10).

To estimate whether our low-depth showcase experiment can be considered representative for larger-scale CASTLING-based experiments, we quantified the dependence of correlation coefficients on the number of compared genes and found that the coefficients of correlation obtained from the complete data sets[38] or an analysis limited to the 435 tagged genes that we had detected correlated well with each other (Pearson's correlation coefficient 0.74; Fig. 5f), indicating a predictive value of our low-depth experiment.

We found that 23 (13%) of 175 tagged genes yielded clearly detectable fluorescence signals in our study, but were not detected by Newman et al. [39] (Fig. 5e, orange points). Since these proteins were also detected by complementary approaches such as mass spectrometry[40] or immunoblotting[4], we assumed that these "false positives" resulted from false-negative clones of the C-GFP library (Fig. 5e). Using independently generated clones based on a different gene tagging strategy[12], we validated the expression of most of these genes when tagged with mNeonGreen, including eight proteins that were not covered in the C-GFP library and neither characterized in any other study analyzed by Ho et al.[38] (Fig. 5g, Supplementary Table 3).

Together, these results highlight the use of CASTLING libraries as a rapid venue for phenotypic profiling and screening experiments when combined with Anchor-Seq to analyze the clone distribution across sub-populations isolated from such libraries.

## Discussion

We developed CASTLING to enable the rapid creation of pooled libraries of clones with large chromosomal insertions such as fluorescent protein tags.

Typically, libraries in yeast have been constructed genewise in an arrayed format using PCR targeting[41]. Based on our own experience[12,42], the construction of arrayed libraries depends on special equipment for parallelization of the procedures, it requires a (costly) resource of arrayed primers for PCR tagging, and handling thousands of strains keeps multiple researchers occupied for several months.

In contrast, fewer resources must be committed to create a library by CASTLING. All the necessary oligonucleotides can be obtained from microarrays, which are about two orders of magnitude more cost-effective than a genome-wide set of conventional solid-phase-synthesized oligonucleotides. Once an established oligonucleotide pool is available, it can be reused to construct a variety of SIC pools containing different features, that is, tags or selection markers. The construction of SIC pools is rapid and can be completed within 1–2 days since the CASTLING workflow avoids preparatory sub-cloning into a plasmid library, which is commonly used in other multiplexed gene-editing approaches[19–21,43]. Transformation and growth of the yeast clones take another 2–3 days, followed by recovery and analysis of the library. This makes library preparation by CASTLING very efficient and therefore it is possible to create a new library for each strain background or mutant of interest. Classical libraries in contrast are confined to the background they were made in and

require genetic crossing to introduce a mutant, which depends on strains specifically constructed for these procedures[44].

In addition to the versatility and flexibility of library creation, tagging fidelity by CASTLING is 90% or higher, exceeding the fidelity observed in conventional gene tagging by PCR targeting, where routinely 50–85% of the obtained clones are correct. It may be worth mentioning that elimination of the false clones during the construction of classical arrayed libraries remains one of the most laborious steps. With CASTLING, false clones cannot obstruct the correct interpretation of a screening because with Anchor-Seq all genotypes can be quantified that are present at the beginning of an experiment as well as their respective enrichment or depletion after phenotypic selection. This allows excluding erroneous genotypes while completing the analysis, which is typically not possible in other multiplexed CRISPR-based gene editing approaches that rely on indirect measures for genotype determination (e.g. sequencing the ectopic crRNA plasmids).

A potential downside of CASTLING and many other pooled library approaches lies within the initial indeterminacy of the exact library composition: Each transformation will yield pools with not exactly the same composition. Currently, genotype coverage with CASTLING can exceed 90% when relatively small libraries with hundreds of genes are created and reproducibly reached 50% for libraries with thousands of genes using <10-fold oversampling (44,000 clones over 5940 ORFs). We have identified that SICs for which the CRISPR target site would be destroyed after integration, or SICs that had a GC-rich crRNA in its PAM-proximal dinucleotides, yielded higher clone numbers as compared to SICs lacking those features (Supplementary Fig. 9).

The identified parameters increased the likelihood of tagging success, but they might also reduce the number of clones for ORFs for which only less efficient SICs could be designed. In this case, additional oversampling would be required. Along this line, a better strategy to increase coverage might be to use successive rounds of CASTLING involving each time a new microarray to target the remainder of genes. The first array would target those genes that can be reproducibly tagged in all trials (Fig. 5b, c), while subsequent arrays would incrementally complete the library with almost proportional scaling efforts in terms of clones to be collected. Probably, it would require 2–4 rounds of CASTLING with a total of 60,000–120,000 clones to tag >60–90% of all 5500–6000 genes in yeast. This would exceed available genome-wide tagging collection, for example, the C-GFP collection[32] with 4159 ORFs (Thermo Fisher), the TAP-tag collection[4] with 4247 ORFs (Dharmacon), or our tandem fluorescent timer collection with 4081 ORFs[42]. Importantly, such an optimization might be necessary only once. Afterwards, all oligonucleotide pools could be used in parallel to generate a nearly complete library. This approach might also yield optimized rule sets to guide the development of CASTLING for a different species.

A major factor that decreased tagging success seemed to be oligonucleotide quality. CASTLING requires long oligonucleotides >100 bp. Even very small error rates and almost perfect coupling efficiencies during oligonucleotide synthesis will give rise to pools that only contain a minor fraction of full-length error-free oligonucleotides. Furthermore, we observed that the same sequences synthesized in different batches gave rise to pools with different performance (pools B1 and B2). We have sequenced and thoroughly analyzed one of the oligonucleotide pools for large library creation. Only a fraction of the designed sequences was represented by perfect full-length oligonucleotides. Most frequently we observe deletions and single-nucleotide polymorphisms (SNPs) in the oligonucleotide sequences. SNPs seem to be more frequent at the 3′ end of the oligonucleotide (which is synthesized first), whereas deletions become more frequent towards the 5′ end of the oligonucleotide (which is

synthesized last). Indeed, error-free synthesis of long oligonucleotides remains challenging[45,46]. To increase the chance of representing each target locus by a perfect oligonucleotide, it might be beneficial to use as many different oligonucleotides per gene as possible or to include multiple redundant sequences.

It is important to stress that faulty oligonucleotides do not necessarily impact the fidelity of the tagging because the in vitro recombineering steps and the in vivo recombination[47] all select against faulty oligonucleotides. Also, errors in the crRNA will most likely render it inactive. Consequently, only a few oligonucleotides that end up in the genome are associated with frameshift errors that impair the expression of the tag. This is impressively demonstrated with the nuclear protein libraries that were prepared with three different oligonucleotide pools, all of which showing >90% in-frame tagging rates (Fig. 3h). This results in intrinsic quality control during CASTLING yielding correctly tagged genes in the majority.

**Prospective applications of CASTLING.** In combination, CASTLING and quantitative Anchor-Seq enable the rapid creation and analysis of pooled libraries with tagged genes. Since each reaction tube contains an entire library, the pooled format is able to address much broader, comparative questions, including different genetic backgrounds and/or environmental conditions.

CASTLING is a method for gene tagging, and the type of screen that can be performed with such libraries entirely depends on the used tag. Therefore, it is up to the creativity of the researcher to develop a screening procedure to convert the information provided by the tags into information about the biological question in mind. Importantly, a screening procedure requires physical fractionation of the library into sub-pools based on a suitable phenotypic read-out, for example, using tags that enable the coupling of a protein behavior such as protein localization[48] or protein–protein interactions[10] with a growth phenotype.

In our opinion, fluorescent protein reporters constitute a particularly attractive group of tags as they provide visual insights into the cellular organization and dynamics, changes of which are associated with many disturbances of biological processes. Our simple FACS enrichment experiment (Fig. 5d–g) can serve but as proof of principle in this regard as current flow cytometry-based cell sorters cannot resolve complex cellular phenotypes, such as the subcellular localization of proteins[49]. We think that for methods such as the recently developed image-activated cell sorting[23], CASTLING can enable a variety of entirely new experimental designs and analyses, ranging from functional genomics to biomedical research, paving the way to a new paradigm of shot-gun cell biology.

Beyond yeast, CASTLING could be adapted for other organisms able to repair DNA lesions by homologous recombination, including bacteria, fungi, flies, and worms, and potentially also in plants and mammalian cells. First evidence that this is the case is provided in the pre-print from Fueller et al. [50], where we show that an adapted SIC strategy can be used for efficient endogenous tagging of genes in mammalian cells. We have preliminary data suggesting that CASTLING also works in mammalian cells, although the size of the library that can be generated with it is currently unclear.

Please note that inadequate adoption of CASTLING can unwittingly generate clones qualified to initiate a gene-drive upon sexual reproduction[51,52]. This can be easily prevented (Supplementary Note 2).

In summary, our work shows that CASTLING libraries and quantitative genotype analysis using Anchor-Seq seamlessly integrate into existing (and upcoming) high-throughput cell sorting instrumentation to enable functional analyses of pooled resources. This outlines new avenues for the investigation of complex cellular processes in direct competition with strategies based on arrayed library resources.

## Methods

**Yeast strains and plasmids.** All strains were derived from ESM356-1 (*S. cerevisiae* S288C, *MAT*a *ura3-52 leu2*Δ1 *his3*Δ200 *trp1*Δ63, which is a spore from strain FY1679[13,53]) and are listed in Supplementary Table 4. Plasmids are listed in Supplementary Table 5. Human codon-optimized Cas12a (formerly Cpf1) family proteins[24] of FnCas12a, *Lachnospiriceae bacterium ND2006* (LbCas12a), *Acidaminococcus* sp. *BV3L6* (AsCas12a), and *Moraxella bovoculi 237* (MbCas12a) were expressed using the galactose-inducible *GAL1* promoter[54] from plasmids integrated into the *ura3-52* locus (pMaM486, pMaM487, pMaM488, pMaM489).

**Cell lysis and Western blot detection of HA-tagged proteins.** Denaturing protein extracts from yeast cells were prepared using incubation with NaOH/β-mercaptoethanol followed by precipitation with trichloroacetic acid and protein solubilization with 6 M urea containing sample buffer for sodium dodecyl sulfate -polyacrylamide gel electrophoresis[5]. Proteins were resolved on Tris-glycine-buffered 10% (v/v) polyacrylamide gels by electrophoresis at 200 V for 90 min, transferred onto a nitrocellulose membrane by wet blotting (12 mM Tris, 96 mM glycine, 20% (v/v) methanol) at 25 V for 120 min, blocked with 10% (w/v) milk powder in blotting buffer (20 mM Tris, 150 mM NaCl, 0.1% (w/v) Tween-20), and the proteins of interest detected with monoclonal mouse anti-Pgk1 (R & D Systems, Fisher Scientific, 1:2,500) and monoclonal mouse anti-HA (12CA5, Sigma-Aldrich, 1:2,000) antibodies bound in 5% (w/v) milk powder in blotting buffer at 4 °C overnight. The surplus of unbound primary antibody was washed away and the secondary horse radish peroxidase-coupled antibody (1:10,000) applied in 5% (w/v) milk powder in blotting buffer at room temperature.

**CASTLING library design.** To facilitate oligonucleotide design, an *R* package (castR) is available from our repository (https://github.com/knoplab/castR/tree/v1.0) that ships along with a graphical user interface (GUI). For small genomes, the GUI can be accessed online (http://schapb.zmbh.uni-heidelberg.de/users/knoplab/castR/). The principles used for oligonucleotide design are described in Supplementary Note 1.

Oligonucleotide sequences used for microarray synthesis of oligopools in this study are given in Supplementary Data 1 (for arrays used in Fig. 3), Supplementary Data 2 (for the array used in Fig. 4), and Supplementary Data 3 (for the array used in Fig. 5).

**Generating SICs for individual genes.** Individual SICs were generated by PCR using a corresponding plasmid template (Supplementary Table 5) and using primers (Supplementary Table 6) that introduced the required 5′ and 3′ homology arms along with a locus-specific crRNA spacer. Cycling conditions for VELOCITY DNA polymerase-based amplification (Bioline) were 97 °C for 3 min, followed by 30 cycles of 97 °C (30 s), 63 °C (30 s), 72 °C (2 min 30 s), and a final 72 °C (5 min) extension hold. The reactions were column purified and adjusted to equal SIC concentration before yeast cell transformation.

**Amplifying oligonucleotide pools and feature cassettes.** The oligonucleotide pools used in this study (Supplementary Table 7) were synthesized by either CustomArray Inc. (pools A and C), Twist Bioscience (pools B1 and B2), or Agilent Technologies (pool D), and reconstituted in TE in case they arrived lyophilized. Pool dilution and annealing temperature were optimized in each case to yield a uniform product of the expected length (Supplementary Fig. 5a, Supplementary Table 1–2). In this study, pool C was diluted 1000-fold and 1.5 fmol were amplified using VELOCITY DNA polymerase (Bioline) with forward primer pool-FP1 and reverse primer pool-RP2 using the following PCR conditions: 97 °C for 3 min, followed by 20 cycles of 97 °C (30 s), 58 °C (30 s), 72 °C (20 s), and a final 72 °C (5 min) extension hold. To keep library member representation as uniform as possible, using more input material and higher annealing temperatures is desirable, as this will usually require fewer PCR cycles for amplification of the full-length synthesis product. All other pools were designed to allow for amplification in 15 cycles using Herculase II DNA polymerase (Agilent Technologies) with forward primer pool-FP2 (or pool-FP3, as indicated) and reverse primers pool-RP2 (or pool-RP3). Cycling conditions were: 95 °C for 2 min, followed by six cycles of 95 °C (20 s), touch down from 67 °C (20 s, $\Delta T = -1$ °C per cycle), 75 °C (30 s), then nine cycles of 95 °C (20 s), 67 °C (20 s), 72 °C (30 s), and a final 72 °C (5 min) extension hold. Primers and truncated oligonucleotides (<75 bp) were removed using NucleoSpin Gel and PCR clean-up columns (Machery-Nagel GmbH & Co. KG). Feature cassettes were amplified by PCR using cognate cassette-FP and cassette-RP and any compatible plasmid template (50 ng, Supplementary Table 5) under the following conditions: 97 °C for 3 min, followed by 30 cycles of 97 °C (30 s), 63 °C (30 s), 72 °C (2 min 30 s), and a final 72 °C (5 min) extension hold. The reaction was treated with *Dpn*I (New England Biolabs) in situ and cleaned-up using NucleoSpin Gel and PCR clean-up columns. For PCR, VELOCITY high-fidelity

DNA polymerase (Bioline) was used with the manufacturer's reaction mix supplemented with 500 µM betaine (Sigma-Aldrich). For analysis, 2 µL of the reaction were used for DNA gel electrophoresis (0.8% or 2.0% agarose in TAE (Tris-acetate-EDTA), Supplementary Figure 5a).

**Recombineering step 1**. Circularization of the amplified oligonucleotide pool (0.8 pmol) with the amplified feature cassette (0.2 pmol) was performed using NEBuilder HiFi DNA Assembly Master Mix (New England Biolabs) in a total reaction volume of 20 µL at 50 °C for 30 min. For analysis by DNA gel electrophoreses, 10 µL of the reaction were used (0.8% agarose in TAE, Supplementary Fig. 5b).

**Recombineering step 2**. To amplify selectively the circular product from step 1, rolling circle amplificytion (RCA) using phi29 was used. First, the annealing mixture was set up (total volume: 5 µL in a PCR tube) using 1 µL of the crude or gel-purified circularization reaction, 2 µL exonuclease-resistant random heptamers (500 µM, Thermo Fisher Scientific), 1 µL of annealing buffer (stock: 400 mM Tris-HCl, 50 mM $MgCl_2$, pH = 8.0), and 1 µL of water. For annealing, the mixture was heated to 94 °C for 3 min and cooled down in thermocycler at 0.5 °C/s to 4 °C. Then, 15 µL amplification mixture were added (consisting of 2.0 µL 10× phi29 reaction buffer, 2.0 µL 100 mM dNTP mix, 0.2 µL 100× bovine serum albumin, 10 mg/mL, and 0.6 µL phi29 DNA polymerase; all from New England Biolabs). Amplification was allowed to proceed for 12–18 h at 30 °C, followed by heat inactivation of the enzymes at 80 °C for 10 min. For analysis by DNA gel electrophoresis (0.8% agarose in TAE), 0.5 µL of this reaction was used (Supplementary Fig. 5c).

**Recombineering step 3**. To release the SICs, 20 U of the restriction enzyme *Bst*XI (New England Biolabs) were added directly to the amplification reaction and the mixture was incubated for 3 h at 37 °C. Typically, such a reaction yielded 10–20 µg of SICs. For DNA gel electrophoresis, 1 µL was used (Supplementary Fig. 5d).

**Estimating recombineering fidelity by NGS**. The oligonucleotide pools were analyzed by NGS (Figs. 3 and 5) after PCR amplification and after recombineering, including UMIs for de-duplication (Supplementary Fig. 7a). For the PCR amplicons, fragments with UMIs were generated using 200 ng starting material (purified by ethanol precipitation) in two cycles of PCR with Herculase II Fusion DNA Polymerase (Agilent Technologies) using an equimolar mixture of P023poolseqNN-primers (1 mM final concentration) in a 25 µL reaction. Cycling conditions were based on the manufacturer's recommendations (62 °C annealing, 30 s elongation). The reactions were purified with NucleoSpin Gel and PCR clean-up columns using diluted NTI buffer (1:5 in water) to facilitate primer depletion, and the fragments eluted in 20 µL 5 mM Tris-HCl (pH = 8.5) each. To remove residual primers, 7 µL of eluate were treated with 0.5 µL exonuclease I (*Escherichia coli*, New England Biolabs) in 1× Herculase II reaction buffer (1 h, 37 °C) and heat inactivated (20 min, 80 °C). The reaction was used without further purification as input for a second PCR (Herculase II Fusion DNA Polymerase, 30 cycles, 72 °C annealing, 30 s elongation) to introduce indexed Illumina-TruSeq-like adapters (primer Ill-ONP-P7-bi7NN and Ill-ONP-P5-bi5NN). The products were size selected on a 3% NuSieve 3:1 Agarose gel (Lonza), purified using NucleoSpin Gel and PCR clean-up columns, and quantified on a Qubit Fluorometer (dsDNA HS Assay Kit, Thermo Fisher Scientific) and by quantitative PCR (qPCR) (NEBNext Library Quant, New England Biolabs, LightCycler 480, Roche). SIC pools were processed likewise using tRNA-seqNN and mNeon-seqNN as primers to introduce UMIs. All samples were pooled according to the designed complexity and sequenced on a NextSeq 550 system (Illumina) with 300 cycle paired-end chemistry.

We sequenced the oligonucleotide pool after PCR amplification, and the SIC pool obtained from the recombineering procedure (Fig. 4). In the latter instance, fragments compatible with Illumina NGS were generated digesting the products of RCA with *Bts*ªI (55 °C, 90 min, New England Biolabs) and *Sal*I-HF (37 °C, 90 min, New England Biolabs). The fragments were column purified, diluted to 100 ng/µL, and blunted using 1 U/µg mung bean nuclease under the appropriate buffer conditions (New England Biolabs). The DNA fragments of 150–200 bp length were gel extracted on 3% NuSieve 3:1 Agarose (Lonza). Both samples were sequenced by GATC Biotech AG (Konstanz, Germany) using Illumina MiSeq 150 paired-end NGS technology.

**Transformation of SICs**. For transformation of individual SICs or SIC pools, Cas12a-family proteins were transiently expressed by making frozen competent cells using either yeasts strains with *GAL1*-controlled Cas12a proteins grown in YP (1% yeast extract + 2% peptone) or SC (synthetic complete) medium containing 2% (w/v) raffinose and 2% (w/v) galactose as carbon source. For transformation[55], the heat shock was extended to 40 min and no dimethyl sulfoxide was added. Recovery of cells that required selection for dominant antibiotic resistance markers (G-418, hygromycin B and clonNAT[56]) was allowed for 5–6 h at room temperature in YP-Raf/Gal (yeast extract peptone dextrose medium containing raffinose and galactose) or YPD (yeast extract peptone dextrose) to proceed prior to plating them on corresponding selection plates.

SIC pools were transformed at a total of 1 µg per 100 µL of frozen competent yeast cells (approximately $2 \times 10^8$ cells). Per library approximately 5 of such transformation reactions were combined corresponding to a yeast culture volume of 50 to 100 mL ($OD_{600} = 1.0$) to generate the competent cells. The number of transformants per library was calculated from serial dilutions. Replica plating on selective plates was used to exclude transiently transformed clones. After outgrowth, libraries were harvested in 15% glycerol and stored at –80 °C. For subsequent experiments, including genotyping, approximately 10,000 cells per clone were inoculated in YPD, diluted to $OD_{600} = 1.0$ (approximately 50 mL of culture), and grown overnight. If necessary, a second dilution was performed to obtain cells in exponential growth phase.

For co-integration experiments using individual SICs, 1 µg DNA per SIC and condition was transformed using 50 µL competent yeast cells. Colony number and fluorescence images were acquired after the sample had been spread onto selective plates. Potential co-integrands were tested by replica plating, streaking, and fluorescence microscopy.

Each transformation mixture was split into two parts containing 1/20 (libA) or 19/20 (libB) of the volume. The largest sample was plated onto four $25 \times 25$ cm$^2$ square plates with YPD + G-418. No replica plating was performed before the libraries were cryo-preserved in 2.5, 10, and 50 mL of 15% glycerol, respectively.

For libraries libC, and the small nuclear library (based on P1), the transformation mixture was plated onto two $25 \times 25$ cm$^2$ plates with YPD + hygromcyin B.

**Fluorescence microscopy**. Cells were inoculated at an $OD_{600} = 0.5$ per condition in 5 mL low-fluorescent SC medium (SC-LoFlo[57]) from cryopreservation stocks and grown overnight, followed by dilution to $OD_{600} = 0.1$ in 20 mL SC-LoFlo the next morning and imaging during mid-exponential growth in the afternoon. Cells were attached to glass-bottom 96-well microscopy plates (MGB096-1-2-LG-L, Matrical) using concanavalin A coating[58]. High-resolution fluorescence micrographs were taken on a Nikon Ti-E epifluorescence microscope equipped with a 60x ApoTIRF oil-immersed objective (1.49 NA, Nikon), a 2048 × 2048 pixel (6.5 µm), an sCMOS camera (Flash4, Hamamatsu), and an autofocus system (Perfect Focus System, Nikon) with either bright field, 469/35 excitation and 525/50 emission filters, or 542/27 excitation and 600/52 emission filters (all from Semrock except 525/50, which was from Chroma). For each condition, a z-stack of 10 planes at 0.5 µm distance was acquired each with a bright field, a short (75% excitation intensity, 10 ms) and a long fluorescence exposure (100% excitation intensity, 100 ms) regimen. For display, the fluorescent image stacks were z-projected for maximum intensity, and cell boundaries taken from out-of-focus bright field images. For imaging cells in Fig. 3 (small nuclear pools), cells were inoculated from cryopreservation stocks and grown overnight in selective synthetic media (SC with monosodium glutamate and hygromycin B). The next morning, the cells were diluted in the same medium and grown to mid-exponential phase. Z-stacks were acquired using 17 planes and 0.3 µm spacing between planes.

**Fluorescence-activated cell sorting**. A homogenous population of small cells (mostly in the G1 phase of the cell cycle) were selected using forward and side scatter. Single cells were sorted according to fluorescence intensity using fluorescence-activated cell sorting performed on a FACSAria III (BD Diagnostics) equipped for the detection of green fluorescent proteins (excitation: 488 nm,; long pass: 502LP,; bandpass: 530/30). We first isolated cells (three million in total), which represented roughly the 30% most fluorescent cells in library #1.1 (Supplementary Table 2) as judged by comparison to cells from strain ESM356-1, which was used as a negative control. The population of fluorescent cells was then grown to exponential phase and sorted into eight fractions (bins) of 125,000 cells each (except for 62,500 cells sorted into bin 8) using bin sizes of roughly 5% (bin 1), 20%, 20%, 20%, 25%, 5%, 5%, 1% (bin 8) according to the $\log_{10}$-transformed intensity of fluorescence emission of small (G1) cells. Sorted pools were grown overnight and the cells were harvested for genomic DNA extraction and target enrichment NGS by Anchor-Seq.

**Library characterization by Anchor-Seq**. To determine cassette integration sites in CASTLING libraries, we used a modified Anchor-Seq protocol:[12] Libraries #1.1, #1.2, and #1.3 (Fig. 4) were prepared with vectorette bubble adapters (vect_illumina-P5 and vect_illumina-P7) that themselves contained barcodes for multiplexing several samples in the same sequencing run. For all other libraries (Figs. 3 and 5), the adapters contained UMIs to account for PCR bias during NGS library preparation (Supplementary Fig. 7b); the barcodes for multiplexing were introduced at the stage of the Illumina sequencing adapters. Genomic DNA (gDNA) was isolated from a saturated overnight culture (approximately $2 \times 10^8$ cells) using YeaStar Genomic DNA Kit (Zymo Research). Genomic DNA (125 µL at 15 ng/µL in ultrapure water) was fragmented by sonication to 800–1000 bp in a microTUBE Snap-Cap AFA Fiber on a Covaris M220 focused ultrasonicator (Covaris Ltd.). In our hands, 51 s shearing time per tube, a peak incident power of 50 W, a duty factor of 7%, and 200 cycles per burst robustly yielded the required size range. Adapters were prepared by combining 50 µM of the respective Watson and Crick oligonucleotides (Supplementary Table 6). Each mixture was heated up to 95 °C for 5 min, followed by cooling to 23 °C in a large water bath over the course of at least

30 min. Annealed adapters were stored at –20 °C until use. We prepared an equimolar mixture of annealed adapters that contained either none, one, or two additional bases inserted after the UMI (halfY-Rd2-Watson and halfY-Rd2-NN-Crick) to increase heterogeneity of the sequencing library. The fragmented genomic DNA (55.5 μL) were end repaired and dA tailed (NEBNext Ultra End Repair/dA-Tailing Module, New England Biolabs) and ligated to 1.5 μL of the 25 μM annealed adapter mix (NEBNext Ultra Ligation Module, New England Biolabs). Products larger than 400 bp were purified by gel excision (using NuSieve, described above) and eluted in 50 μL 5 mM Tris-HCl (pH = 8.5). SIC integration sites were enriched by PCR (NEBNext Ultra Q5 Master Mix, New England Biolabs) using 12 μL of the eluate with suitable pairs of adapter- and SIC-specific primers. Initial denaturation was 98 °C (30 s), followed by 15 cycles of 98 °C (10 s), and 68 °C (75 s). Final extension was carried out at 65 °C (5 min). Reactions were purified using Agen-court AMPure XP beads (0.9 vol, Beckman Coulter). The fragments were further enriched in a second PCR using the custom-designed primers Ill-ONP-P7-bi7NN and Ill-ONP-P5-bi5NN to introduce technical sequences necessary for multiplexed Illumina sequencing. After size selection by gel extraction (250–600 bp), NGS library concentrations were measured by Qubit Fluorometer (dsDNA HS Assay Kit, Thermo Fisher Scientific) and by qPCR (NEBNext Library Quant, New England Biolabs, LightCycler 480, Roche). Furthermore, their size distribution was verified either on a Fragment Analyzer (Advanced Analytical Technologies Inc) or by gel electrophoresis of the qPCR product. Quantified libraries were sequenced on a NextSeq 500 (for pool C, Deep Sequencing Core Facility) or on a NextSeq 550 sequencing system (both Illumina, 300 cycle paired end). If necessary, 10–15% phiX gDNA was spiked in to increase sequence complexity.

For MinION nanopore sequencing, the first PCR was carried out as described above for library #1.1 (using 20 cycles) to introduce barcodes for multiplexing FACS bins on the same sequencing run, column purified, and the NGS library was prepared for 1D sequencing by ligation (SQK-LSK108) according to the manufacturer's protocols (Oxford Nanopore Technologies). Sequencing was performed on a MinION device using R9.4 chemistry (Oxford Nanopore Technologies). Samples were multiplexed considering the number of different clones present in a pool, bin size, gDNA yield after extraction, and yield of the first PCR.

**Insertion junction sequencing of non-fluorescent cells.** Cells from library 1a were grown in selective synthetic media (SC with monosodium glutamate and hygromycin B) for approximately eight generations, and non-fluorescent cells were sorted into glass-bottom 384-microscopy plates using a FACS Aria III as described under "FACS". The absence of fluorescence was confirmed by fluorescence microscopy and 60 non-fluorescent clones were pooled and grown overnight to full density. Anchor-Seq amplicons were prepared as described under "Library characterization by Anchor-Seq" using primers NegCells-NNN (Supplementary Table 6). The amplicons were size selected (~600 bp) and cloned using the NEB PCR Cloning Kit (New England Biolabs). The resulting amplicons were Sanger sequenced at Eurofins Genomics (Cologne, Germany).

**Illumina NGS data analysis and read counting.** Raw reads (150 bp paired-end) were trimmed and de-multiplexed using a custom script written in Julia v0.6.0 with BioSequences v0.8.0 (https://github.com/BioJulia/BioSequences.jl). Read pairs were retained upon detection of basic Anchor-Seq adapter features. Next, these reads were aligned to a reference with all targeted loci using bowtie2[59] v2.3.3.1. Such references comprised the constant sequence starting from the feature cassette amplified by PCR and 600 bp of the respective proximal genomic sequence of *S. cerevisiae* strain S288C (R64-2-1). For off-target analysis, the constant Anchor-Seq adapter features were trimmed off the reads. The remaining variable sequence of the reads was then aligned with bowtie2[10] to the complete and unmodified genome sequence of *S. cerevisiae* strain S288C (R64-2-1). A read pair that aligned to the reference was counted if both reads of the pair were aligned, such that the forward read started at the constant region of the Anchor-Seq adapter-specific primers. In addition, we set the requirement that the inferred insert size was longer than the sequence provided for homologous recombination during the tagging reaction. Counting was implemented using a custom script (Python v3.6.3 with HTSeq 0.9.1[60] and pysam 0.13). In case UMIs were included in the Anchor-Seq adapter design, they were normalized for sequencing errors using UMI-tools (version 0.5.3)[61].

For analysis of data obtained from amplicon sequencing (i.e., from PCR and SIC amplification reactions), the reads were either denoised from sequencing errors using dada2 (version 1.5.2)[37] to evaluate fidelity and abundance or directly aligned with bowtie2 to a reference build from the designed oligonucleotides. Denoised reads were assigned to loci based on the minimal hamming distance to designed oligonucleotides.

**Analysis of nanopore sequencing data and read counting.** Nanopore sequencing yields very long reads. Therefore, the reference was assembled as aforementioned but using 2000 bp of the locus-specific sequences plus the constant sequence of the cassette enriched by the Anhor-Seq reaction. MinION data were basecalled using the Albacore Sequencing Pipeline Software v2.0.2 (Oxford Nanopore Technologies). For data analysis, a custom script was used to extract and de-multiplex informative sequence segments from all reads based on approximate matching of amplicon features (e.g., the constant region of the vectorette or feature cassette;

Julia v0.6.0 with BioSequences v0.8.0, see above). Matching with a Levenshtein distance of 1 was sufficient to discriminate between the barcodes used in this study. Then, the extracted sequence segments were aligned to the reference using mini-map2 (v2.2-r409)[62], using the default parameters (command line option: "-ax map-ont") for mapping of long noisy genomic reads. Only reads that mapped to the beginning of the reference were counted using a custom shell script. The count data for the clones retrieved in each library for cells contained in the individual bins after FACS are provided in Supplementary Table 3.

**Calculation of fluorescence intensity estimates.** Fluorescence intensity estimates were calculated as previously described for FACS-based profiling of pooled yeast libraries:[63] Let $b$ be a natural number from 1 to 8 indicating one of our eight FACS bins $B$ for which we know the fraction $p_b$ of the total cell population sorted into this bin. Further, we determined by sequencing for each bin the number of reads $r_{g,b}$ of an individual genotype $g$ (tagged ORF) of all detected genotypes $G$. The observed unnormalized cell distribution of $g$ is given by:

$$\tilde{C}_g(b) = \frac{r_{g,b}}{\sum_{g \in G} r_{g,b}} p_b. \tag{1}$$

We define the fluorescence intensity estimate for $g$ as the empirical mean of $\tilde{C}_g$:

$$\text{fluorescence intensity estimate} := E_{f \sim \tilde{C}_g}[b] = \sum_b \frac{b \cdot \tilde{C}_g}{\sum_{b \in B} \tilde{C}_g}. \tag{2}$$

**Calculations and statistical analyses.** Statistical analyses were performed using $R$ as specified in the scripts or legends.

**Estimation of co-integrand number.** We assumed that most co-integrands would result from doubly transformed individuals. So, the number of phenotypic heterozygous individuals (e.g., GFP + RFP + or kan$^R$ hyg$^R$) represents half of the co-integrands if both feature cassettes that were transformed at equimolar ratios have an equal probability of being taken up with the likes of them (i.e., GFP+ GFP+ and RFP+ RFP+) as with each other. Further, we assumed that the fluorescent protein or the antibiotic resistance marker present in the feature cassette had no or only a minor impact on integration efficiency.

**Calculation of copy number changes during RCA.** Copy numbers (UMI counts) were normalized to the median UMI frequency in each sequencing experiment and the Gaussian kernel density estimate plotted. Fold changes were calculated as normalized UMI counts after RCA divided by normalized UMI counts after PCR for each oligonucleotide.

**Software and figure generation.** Proportional Venn diagrams were generated using eulerAPE[64]. Analyses were performed using R v3.4.1/v3.5.1 with Biostrings v2.44.2[65] and data.table v1.10.4/v1.11.4. Plots were generated using ggplot2 v2.3.0 and figures were made using Apple Keynote 8.2.

## Data availability

Raw sequencing data has been deposited at the BioProject database under accession code PRJNA545279 as well as at heiDATA (https://doi.org/10.11588/data/L45TRX). Plasmids and plasmid maps are available upon request. The source data underlying Figs. 1b–d, 3b–i, 4a–c, 5b, c, e and Supplementary Figs. 1, 2, 3a–c, 4, 5a–d, 8, 9, and 10 are provided as Source Data file. Any other relevant data is available from the authors upon request.

## Code availability

The source code of the *R* shiny application for oligonucleotide design is available from our github repository (https://github.com/knoplab/castR/tree/v1.0).

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

## Acknowledgements

The authors wish to thank Ilia Kats, Cyril Mongis, and Krisztina Gubicza for help with IT infrastructure and experiments. We acknowledge support from the Deutsche Forschungsgemeinschaft (DFG KN498/12–1), the state of Baden-Württemberg through bwHPC for high-performance computing and SDS@hd for data storage (grant INST 35/1314–1 FUGG), and the Dietmar Hopp foundation. K.H. was supported by a HBIGS graduate school fellowship. We also acknowledge help from the Flow Cytometry Core Facility at ZMBH, and the Deep Sequencing Core Facility of the University of Heidelberg, both of which are supported by the CellNetworks cluster of excellence. E.D.L. acknowledges support from A.-M. Boucher, from the Estelle Funk Foundation, the Estate of Fannie Sherr, the Estate of Albert Delighter, the Merle S. Cahn Foundation, Mrs. Mildred S. Gosden, the Estate of Elizabeth Wachsman, the Arnold Bortman Family Foundation. E.D.L. is incumbent of the Recanati Career Development Chair of Cancer Research.

## Author contributions

M.K. conceived the project. M.K., B.C.B., K.H., and M.M. designed the experiments and B.C.B., K.H., M.M., and D.K. performed the experiments. E.D.L. and E.S. contributed methods. K.H., B.C.B., M.K., and M.M. analyzed the data. M.K. and B.C.B. wrote the manuscript. All authors read and approved the final manuscript.

## Additional information

**Competing interests:** The authors declare no competing interests.

