## [Peer Review File · Nature Communications]

Reviewers' Comments:

Reviewer #1:

Remarks to the Author:

This submission by Buchmuller and colleagues presents a technique for generating large-scale, pooled collections of tagged strains. The technique is based on synthesis of a variable oligonucleotide that includes a CRISPR guide and the homology arms for genomic regions surrounding the intended cut site. These oligo pools are then used as primers against a common, larger insertion cassette (containing, for example, GFP), and after assembly, the result is a collection of homologous repair templates that can each insert into a unique place in the genome, such as at the C'terminus of target proteins.

The molecular biology of this study appears sound, and this will be a useful way to construct pooled libraries of yeast cells. However, I believe this paper can be further strengthened prior to publication.

Major comments:

1) The main limitation of this manuscript is that the demonstrated use case for a pool of variously-tagged yeast cells is underwhelming. limited to the identification of 8 genes for which C'terminal GFP tags lead to varying levels of brightness. That this technique really works start-to-finish would be better demonstrated by assays for which there are known positive controls, in order to estimate both false negative and false positive rates. If these pools cannot be effectively screened, then their utility is greatly diminished.

2) Is the use of these libraries limited to the AnchorSeq technique, and Oxford Nanopore readouts? To be more widespread in its utility, could the pools be quantitated via the crRNA sequence itself (which would presumably serve as a barcode) and then quantitated on other sequencing platforms?

Minor comments:

1) As an active consumer of oligonucleotide pools, it would be helpful to see more comparisons between the different oligonucleotide providers. The nomenclature between the text and the methods was inconsistent, and no overall summary / recommendation was given. It would be helpful if that was added.

2) Line 120: "We detected only few individual colonies where both genes were fluorescently tagged (Fig. 1c), independent of the relative concentration of the two SICs used for transformation (Fig. 1d)." The authors should note that, presumably, this means that most cells receive only 1 HR template. This would not be the case were they to use a similar technique in mammalian cells (as they mention in the discussion) via transfection, as in that case each cell that takes up plasmid DNA would receive many different molecules from the pool.

3) Supp. Fig. 6c The diagram is confusing. What do the nt numbers refer to? The linkers? The schematized parts? Also, "CIRSPR" typo (twice).

Reviewer #2:

Remarks to the Author:

In the manuscript by Buchmiller et al., the authors present a method, termed CASTLING, which aims to enable the creation of yeast clone collections with gene sets of interest tagged with effector proteins of interest (eg GFP). To do so, the authors devised a scheme using self-integrating cassettes to simultaneously tag loci of interest and enable scalable readouts. The clever part of the method is that locus-specific elements are synthesized in a pooled format and the universal elements are subsequently cloned in, amplified, and introduced into yeast such that

single yeast generally only get one modification. The authors demonstrate their approach by constructing several custom yeast clone collections and thoroughly characterizing the molecular pools along the way. This reviewer appreciated the pragmatic language used by the authors to clearly weigh the benefits and drawbacks of their approach as well as describe the technical challenges. While I would like to see an application of their technology, beyond sorting proteins of different expression levels, I can agree that this is beyond the scope of this manuscript. Overall the manuscript is well written and the experiments and data analysis are sound. The approach is exciting and will be a valuable tool for the community.

Specific points:

1. Page 1 line 23-24: It's unclear what the authors mean by "~10-fold oversampling.". This phrasing is repeated in the discussion and could be explained more precisely.
2. Page 2 line 50-51: CRISPR-associated, not CRISPR.
3. All usage of Cpf1 should be converted to Cas12a as this is the appropriate nomenclature. The authors can point out in the beginning that Cas12a was formerly known as Cpf1.
4. –The figures, while aesthetically pleasing, are often not directly labeled and it is therefore unclear what is being presented without carefully reading the figure legends. I strongly encourage the authors to directly label their axes with values/units to enhance clarity.
5. In Figure 3b-c, a more intuitive way to present the data would be in terms of percent of the library (b) and genotypes (c).
6. It was challenging to ascertain exactly how the UMIs were implemented, even after consulting the text, methods, figure legends, and a prior publication. The authors should provide a schematic clearly depicting their steps for NGS library prep.
7. The others point out that one of their libraries was contaminated. Why was this included and how do we know other libraries are not contaminated. This would be observable if it was contaminated by one specific oligo not in the synthesis tool, but not observable if contaminated by other libraries.
8. The ultimate success rate for generating target ORF collections was concerningly low, which the authors claim is due to oligo synthesis and not the CASTLING methodology. In particular, the authors were only able to retrieve 57% of the designed oligonucleotides after PCR, 31% of which were error free (Page 7 lines 234-236). That number drops to 51% in the SIC pool, which suggest the errors are occurring during or before the PCR step. The final library only covered 45% of their target ORFs. The error rates that the authors observed far exceeds the error rates observed in oligo pools produced by the companies used by the authors (Twist, Custom Array, Agilent). Thus, either something about their oligo design results in reduced synthesis fidelity or something is wrong with their PCR amplification and deep sequencing protocols. Consequently, I would like the authors to elaborate on their process and results further so I, and the readers, can better evaluate whether it was due to synthesis or their methods. For example, what QC did the companies perform/guarantee, how often do you see different mutations associated with the same UMI, what are the character of mutations you observe (eg mutations or indels), and so on. Lastly, as there is some discrepancy in naming the pools in the figures and the methods the authors should update these to be concordant.
9. Page 8 Line 244, Figure 4c and the raw values suggest that 20% is an error and should be replaced by 34%.
10. The authors speculate on the gene editing outcomes in in non-fluorescent cells (Figure 3g-h

and Page 6 lines 199-). I suggest that the authors collect these non-fluorescent cells and unambiguously describe the genotypes to ensure systematic misinterpretations are not made.

11. The authors should elaborate on the potential impacts of a 0.2% non-specific integration rate. What impact would this have on future applications? What can be done to prevent this? Etc.

CASTLING Manuscript: 36766

In addition to the changes introduced based on reviewer comments we made some additional minor changes:

- line 114: “Supplementary Fig. 3” to “Supplementary Fig. 3a–c”
- line 146: “oligos” to “oligonucleotides”
- line 168 “Supplementary Figure 7a” to “Supplementary Fig. 7a”
- line 171 “Supplementary Figure 7b” to “Supplementary Fig. 7b”
- line 171: “along with the sequence adjacent” to “along with the genomic sequence adjacent”
- line 186: “2 fold” to “2-fold”
- line 186: “used starting material and recovery” to “starting material and its recovery”
- line 247: “sub-cellular” to “subcellular”
- line 285: “Supplementary Fig. 9a–b”
- line 292: “subpopulation” to “sub-population”
- line 293: “by FACS fluorescent cells from a library by FACS and” to “by FACS from a library and”
- line 324: “classical libraries, in contrast,” to “classical libraries in contrast,”
- We updated the section on Fluorescence microscopy in the Materials & Methods section

Reviewer #1 (Remarks to the Author):

[...]

Major comments:

1) The main limitation of this manuscript is that the demonstrated use case for a pool of variously-tagged yeast cells is underwhelming. limited to the identification of 8 genes for which C-terminal GFP tags lead to varying levels of brightness. That this technique really works start-to-finish would be better demonstrated by assays for which there are known positive controls, in order to estimate both false negative and false positive rates. If these pools cannot be effectively screened, then their utility is greatly diminished.

Answer:

As we point out in the manuscript, the technology to screen mixed populations based on localization (image activated cell sorting) is not yet broadly available, and hence the chosen example using intensity as a cell sorting criterion is of limited interest. However, to demonstrate that ‘the technique really works start-to-finish’ we performed the experiment shown in Figure 3 where we created a library with CASTLING, in which 215 abundant nuclear proteins are fluorescently tagged (Figure 3). The known and *a priori* validated sub-cellular localization and expression level of these genes in yeast serves as the required positive control, which puts us in the position to estimate all critical parameters. In our opinion, the results of this experiment are excellent proof that CASTLING already works start-to-finish for small libraries and we scrutinize its applicability for creation and screening of larger libraries in various aspects.

Changes in the Manuscript: We have clarified in line 435 “our simple FACS experiment can serve *but* as a proof of principle” to stress the limitation.

2) Is the use of these libraries limited to the AnchorSeq technique, and Oxford Nanopore readouts? To be more widespread in its utility, could the pools be quantitated via the crRNA sequence itself (which would presumably serve as a barcode) and then quantitated on other sequencing platforms?

Answer: Any next-generation sequencing method that provides reads 150 nt or longer is suitable to analyze the libraries. In fact, we mostly use Illumina NGS (Materials and Methods). Oxford Nanopore sequencing was used to showcase rapid on-site analyses (lines 294–295).

We recommend analyzing the libraries using Anchor-Seq. If only the crRNA sequences were quantified by NGS, off-target integration of the SIC could not be distinguished from the correctly tagged (true positive) clones. By using Anchor-Seq, not only the crRNA sequence is analyzed, but also the genomic sequences in the immediate vicinity of the integration site, including the insertion junction. Together, this is a unique constellation that allows to reject irrelevant reads from further analysis. Given the fact that a sequencing run is quite expensive, we think that this justifies the additional steps required to prepare the sample for Anchor-Seq.

Changes in the Manuscript: Because of this discussion here and reviewer #2, point 6, we now incorporated also a schematic visualization of the sequencing work flow (new Supplementary Figure 7).

Minor comments:

1) As an active consumer of oligonucleotide pools, it would be helpful to see more comparisons between the different oligonucleotide providers. The nomenclature between the text and the methods was inconsistent, and no overall summary / recommendation was given. It would be helpful if that was added.

Answer: We thank the reviewer for noting that the nomenclature of the oligonucleotide pools in the main manuscript and the Supplementary Information was inconsistent.

In this study, we have used two identical pools (pool B1 and pool B2) synthesized by the same company (TWIST) at different time points. They showed larger differences in their composition (Figure 3a–e), which directly indicates that the quality of the material is batch-dependent. It is therefore not possible to provide general recommendations. Moreover, DNA synthesis technologies are evolving on a fast pace and any recommendation might be no longer valid at some future date this paper is read.

Changes in the Manuscript: We have adjusted the nomenclature in the Materials & Methods and the SI to match the manuscript.

2) Line 120: “We detected only few individual colonies where both genes were fluorescently tagged (Fig. 1c), independent of the relative concentration of the two SICs used for transformation (Fig. 1d).” The authors should note that, presumably, this means that most cells receive only 1 HR template.

Answer: It is not possible to conclude that most cells receive one SIC, because the probability by which a HR template that is inside the cell is integrated into the genome is not known. For this work, this is however irrelevant. The only parameter that matters is the frequency at which two different SICs are integrated into the same genome. Our experiments demonstrate this rate is indeed low.

This would not be the case were they to use a similar technique in mammalian cells (as they mention in the discussion) via transfection, as in that case each cell that takes up plasmid DNA would receive many different molecules from the pool.

Answer: In mammalian cells, the double integration rate is very high, as shown in our 2nd manuscript on bioRxiv. However, we have preliminary evidence that transfections in mammalian cells can be optimized to significantly reduce double and multiple integrations.

Changes in the Manuscript: This is a speculative discussion that does not add anything. We therefore did not change the manuscript.

3) Supp. Fig. 6c The diagram is confusing. What do the nt numbers refer to? The linkers? The schematized parts? Also, “CIRSPR” typo (twice).

Answer: Indeed. We have remade this panel. Now it should be much clearer.

Changes in the Manuscript: We have re-designed Supplementary Figure 6c.

Reviewer #2 (Remarks to the Author):

[...]

Specific points:

1. Page 1 line 23-24: It's unclear what the authors mean by “~10-fold oversampling.”. This phrasing is repeated in the discussion and could be explained more precisely.

Answer: The term ‘oversampling’ is routinely used in the context of clone library generation and refers to the number of colonies created or analyzed with respect to the theoretical library diversity. For example, if 200 genes should be tagged and 2,000 colonies were collected from the transformation plates, the library was 10-fold oversampled.

Changes in the Manuscript: We have added the exact numbers: **Line 342** “... using less than 10-fold oversampling (44,000 clones over 5,940 ORFs).”

2. Page 2 line 50-51: CRISPR-associated, not CRISPR.

Answer: corrected.

Changes in the Manuscript: We have added the word ‘associated’.

3. All usage of Cpf1 should be converted to Cas12a as this is the appropriate nomenclature. The authors can point out in the beginning that Cas12a was formerly known as Cpf1.

Answer: We agree.

Changes in the Manuscript: Throughout the manuscript we now have replaced the usage of “Cpf1” with “Cas12a” also for the specific variants such as FnCas12a. As suggested, we now point out in the introduction that Cas12a was formerly known as Cpf1.

4. The figures, while aesthetically pleasing, are often not directly labeled and it is therefore unclear what is being presented without carefully reading the figure legends. I strongly encourage the authors to directly label their axes with values/units to enhance clarity.

Answer: We agree.

Changes in the Manuscript: We have moved axes titles directly next to the axes in Figures 1b, 1d, 3b and 3c, 3d, 3h, 3i, and 5c, as well as in Supplementary Figure 8b (now: Supplementary Figure 9b). We added a missing color legend in Figure 3i.

We have revised panel titles in Figures 1b, 1d, 1g–l, and 5a–c to allow interpretation of the data without consulting the figure legends.

5. In Figure 3b–c, a more intuitive way to present the data would be in terms of percent of the library (b) and genotypes (c)

Answer: Indeed, the data presented in Figure 3b–c can be directly interpreted in terms of percentages.

Changes in the Manuscript: To account for this and for our belief that it is still useful to display the absolute numbers, we added secondary axes in Figures 3b–c indicating percentages.

6. It was challenging to ascertain exactly how the UMIs were implemented, even after consulting the text, methods, figure legends, and a prior publication. The authors should provide a schematic clearly depicting their steps for NGS library prep.

Answer: Thank you for pointing this out. We now have added another figure to outline the strategy.

Changes in the Manuscript: We have added a new Supplementary Figure (Supplementary Figure 7).

7. The others point out that one of their libraries was contaminated. Why was this included and how do we know other libraries are not contaminated. This would be observable if it was contaminated by one specific oligo not in the synthesis tool, but not observable if contaminated by other libraries.

Answer: We believe that including also the contaminations is valuable information. In a real experiment, contaminations are easy to handle, because they can be simply excluded based on the unambiguity of Anchor-Seq results.

This, we outlined in the discussion (line 331): “With CASTLING, ‘false clones’ cannot obstruct the correct interpretation of a screening because Anchor-Seq can analyze, verify, and quantify all genotypes available at the beginning of an experiment as well as their respective enrichment or depletion after phenotypic selection. This allows excluding faulty genotypes while completing the analysis, which is typically not possible in other multiplexed CRISPR-based gene editing approaches which rely on indirect measures for genotype determination.”

In the other experiments we did not observe contaminations.

Changes in the Manuscript: No changes in the text.

8. The ultimate success rate for generating target ORF collections was concerningly low, which the authors claim is due to oligo synthesis and not the CASTLING methodology.

In particular, the authors were only able to retrieve 57% of the designed oligonucleotides after PCR, 31% of which were error free (Page 7 lines 234-236). That number drops to 51% in the SIC pool, which suggests the errors are occurring during or before the PCR step. The final library only covered 45% of their target ORFs.

The error rates that the authors observed far exceeds the error rates observed in oligo pools produced by the companies used by the authors (Twist, Custom Array, Agilent). Thus, either something about their oligo design results in reduced synthesis fidelity or something is wrong with their PCR amplification and deep sequencing protocols. Consequently, I would like the authors to elaborate on their process and results further so I, and the readers, can better evaluate whether it was due to synthesis or their methods. For example, what QC did the

companies perform/guarantee, how often do you see different mutations associated with the same UMI, what are the character of mutations you observe (eg mutations or indels), and so on.

Answer:

The error rates we observe are fully consistent with the error rates for long oligonucleotides (160–170-mers). The synthesis error rates of the oligos arrays we used were about 0.5–1.0% per nt and the coupling efficiencies about 98.0–99.0% per nt, which results in only 2–20 mass% full length error free product.

The companies did not perform QC of the material that we obtained. They are usually very helpful with trouble shooting in case PCR amplification is difficult. This was the case for the first TWIST pool (Pool B1) and we obtained a second identical pool for free. This second pool (Pool B2) performed much better, and it required less PCR cycles for amplification indicating a much higher amount of full length oligos (Figure 3a–e). This suggests batch to batch variability (see also our comment to reviewer #1 Minor Point 1).

At each stage of the molecular recombineering procedure, the erroneous products compete with the full-length perfect products, which is the more of a challenge, the higher the diversity of the library is. As we discuss in the manuscript in details, oligonucleotides with errors in a specific sequence are mostly eliminated in one of the in vitro or in vivo recombineering or recombination steps, eventually giving rise to libraries where each individual clone is correct with > 90–95% probability. This intrinsic quality control is likely to contribute to the drop in coverage (57% to 51%) the reviewer mentions.

The reviewer proposes that we could quantify the frequency of different mutations associated with the same UMI. In the new Supplementary Fig 7 we now outline the sequencing workflow. From this it should be apparent that different ‘mutations’ associated with the same UMI must be PCR or sequencing errors. As mentioned in the manuscript, we used a de-noising strategy (Callahan et al., 2016) to remove these artefacts.

We can answer the reviewer’s question about the type of mutations that are present in the oligonucleotides: Most frequently we observe deletions and SNPs in the oligonucleotide sequences (please see below, point 10). SNPs seem to be more frequent at the 3’ end of the oligonucleotide (which is synthesized first), whereas deletions become more frequent towards the 5’ end of the oligonucleotide (which is synthesized last). We observe a small fraction (< 1%) of mutations appearing in the overlap regions during isothermal assembly, likely caused by the limited fidelity of the enzymes.

In the paper, we have discussed the impact of erroneous oligonucleotides (and/or errors introduced during the recombineering procedure) rather extensively and we propose different strategies to address issues with library complexity, especially in the case of large libraries (lines 348–360).

Changes in the Manuscript: This is a lengthy discussion of the oligo pools and different tech. details. We have added text into the discussion to reflect some of the points mentioned here.

Line 361 “Despite this possibility, a major factor that operated on the decrease in tagging success seemed to be oligonucleotide quality. CASTLING requires long oligos >>100 bp, and even very small error rates and almost perfect coupling efficiencies will give rise to pools that only contains a minor fraction of full-length error free oligos. Furthermore, we observed that the same sequences synthesized on different days give rise to pools with different performance (Pool B1 and B2). We have sequenced and thoroughly analyzed one of the oligonucleotide pools for large library creation. Only a fraction of the designed sequences was represented by full-length perfect oligonucleotides. Most frequently we observe deletions and SNPs in the oligonucleotide sequences. SNPs seem to be more frequent at the 3’ end of the oligonucleotide (which is synthesized first), whereas deletions become more frequent towards the 5’ end of the oligonucleotide (which is synthesized last). Indeed, error-free synthesis of long oligonucleotides remains challenging^{42,43}. To increase the chance of representing each target locus by a perfect oligonucleotide, it might be beneficial to use as many different oligonucleotides per gene as possible or to include multiple redundant sequences (depending on the size of the microarray) if an adequate SIC design for a specific locus is known.

It is important to stress that faulty oligonucleotides do not impact the fidelity of the tagging because the in vitro recombineering steps and the in vivo recombination steps all select against faulty oligos. Also errors in the crRNA will render it inactive. As a consequence, only a few % of the oligos that end up in the genome are faulty, i.e. associated with frameshift errors that impair the expression of the tag. This is impressively demonstrated with the nuclear protein libraries that were prepared with three different oligonucleotide pools, all of which showing >90% in-frame tagging rates (Fig. 3). This may be either because faulty oligonucleotides fail to assemble with the feature cassette or because errors in their crRNA and/or the homology arms render the SIC inactive due to reduced

recombination frequencies. This results in an intrinsic 'quality control' during CASTLING yielding correctly tagged genes in the majority."

Lastly, as there is some discrepancy in naming the pools in the figures and the methods the authors should update these to be concordant.

Changes in the Manuscript: We have adjusted the nomenclature in the Materials & Methods and the SI to match the manuscript.

9. Page 8 Line 244, Figure 4c and the raw values suggest that 20% is an error and should be replaced by 34%.

Answer: The percentages in Figure 4c (as well as in Figure 5b) refer to their share in all tagged ORFs excluding the non-tagged ORFs (1,127 ORFs out of 3,262 ORFs). The overall success rate is 1,127 ORFs out of 5,664 ORFs (20%).

Changes in the Manuscript: No changes in the text.

10. The authors speculate on the gene editing outcomes in non-fluorescent cells (Figure 3g-h and Page 6 lines 199-). I suggest that the authors collect these non-fluorescent cells and unambiguously describe the genotypes to ensure systematic misinterpretations are not made.

Answer: Non-fluorescent cells constitute only a minor fraction (less than 4% on average). We have analyzed a pool of 60 non-fluorescent clones by generating Anchor-Seq amplicons for Sanger sequencing (updated information in Materials & Methods). In all cases, we found correctly inserted SICs at one of the loci targeted by the oligonucleotide pool. However, all of them contained frame shifts in the region of the homology arm due to deletions of one or more nucleotides. This causes the GFP to be out of frame. These results demonstrate that the dark clones are mostly cells with correctly targeted SICs, but where an error in the oligo renders the tag inactive.

Changes in the Manuscript: We have added text to explain this data, and we added a corresponding paragraph in Materials & Methods.

Line 198: "For the clones with no fluorescence signal, we suspected either frame-shift mutations in the polypeptide linker (due to faulty oligonucleotides) or in the fluorescent protein reporter (due to limited fidelity of DNA polymerases), or off-target integration of the SIC. Sequencing of the insertion junctions of 16 dark clones revealed small deletions of one or more nucleotides in the homology arms that direct the SICs to the 3'-end of the ORF. Therefore, the majority of dark clones appears to contain correctly targeted SICs. However, due to errors in the sequences that are derived from the oligonucleotides, the GFP is not in frame. "

11. The authors should elaborate on the potential impacts of a 0.2% non-specific integration rate. What impact would this have on future applications? What can be done to prevent this? Etc.

Answer: As we have outline before, Anchor-Seq allows to eliminate faulty/off-target integrations from the analysis. So, they have no impact on future applications and we do not see a need to further reduce this fraction.

Changes in the Manuscript: No changes in the text.

Reviewers' Comments:

Reviewer #1:

Remarks to the Author:

I am disappointed in the author's response to my request for a 'start-to-finish' demonstration of the utility of this technology. They claim that the nuclear localization study provided in Figure 3 represents a screen, and in the rebuttal write:

"The known and a priori validated sub-cellular localization and expression level of these genes in yeast serves as the required positive control, which puts us in the position to estimate all critical parameters."

Actual screens, however, are not comprised entirely of true positives. With the experiments presented, one cannot determine a false positive rate, which is critical for evaluating any screening technology. If CASTLING is truly "a rapid method to tag a large fraction of the yeast genes" (line 305) then I would think that devising a true screen would be relatively trivial.

The absence of a legitimate screen is quite conspicuous, and without it, I think readers will generally take a wait-and-see approach to the method rather than trying it themselves.

Less important, but I reiterate that I think the authors are doing themselves a disservice by not speculating (which is fine to do in a discussion!) on how those who work in mammalian cell systems could think about applying this approach.

Reviewer #2:

Remarks to the Author:

In the revised manuscript by Buchmiller et al., the authors addressed all of my specific points. I have no further concerns and fully support publication of this manuscript.

Answer to the comments of reviewer #1.

Reviewer #1 (Remarks to the Author):

I am disappointed in the author's response to my request for a 'start-to-finish' demonstration of the utility of this technology. They claim that the nuclear localization study provided in Figure 3 represents a screen, and in the rebuttal write:

"The known and a priori validated sub-cellular localization and expression level of these genes in yeast serves as the required positive control, which puts us in the position to estimate all critical parameters."

Actual screens, however, are not comprised entirely of true positives. With the experiments presented, one cannot determine a false positive rate, which is critical for evaluating any screening technology. If CASTLING is truly "a rapid method to tag a large fraction of the yeast genes" (line 305) then I would think that devising a true screen would be relatively trivial.

The absence of a legitimate screen is quite conspicuous, and without it, I think readers will generally take a wait-and-see approach to the method rather than trying it themselves.

Answer:

While we feel sorry that we did not answer the request of the reviewer to her or his satisfaction within the first round of review, we think that the reason for this is a misunderstanding that needs to be constructively clarified.

During the first round, this reviewer noted "[...] that this technique really works start-to-finish would be better demonstrated by assays for which there are known positive controls, in order to estimate both false negative and false positive rates".

Since the main point of our paper is the construction and quantitative assessment of pooled tag libraries, we interpreted this request so that this reviewer is interested in accurate estimates of false negative and false positive rates of the CASTLING technique for library construction. Within this context, true positives are successfully tagged ORFs which were present within the oligopool by design, true negatives are all ORFs not tagged because they are not present in the oligopool design, false positives are tagging events which were not intended during oligopool design (also called "off-targets"), and finally false negatives, which are unsuccessfully tagged ORFs, programmed in the oligopool, but not observed in the final clone library. To obtain reliable estimates of these parameters, it is required to construct a CASTLING library for which reference data is available for quantitative comparison. This is exactly what we achieve with the experiments shown in Figure 3 and the reason why we directed this reviewer's attention to Figure 3. The experiment is based on an assay (fluorescence microscopy) with unambiguously scorable phenotype (nuclear localization), and hence fulfills this reviewer's initial request for "assays for which there are known positive controls".

In the reply of this reviewer to our answer this reviewer now asks for a *screen* to estimate false negative and false positive rates: "Actual screens, however, are not comprised entirely of true positives. With the experiments presented, one cannot determine a false positive rate, which is critical for evaluating any screening technology. If CASTLING is truly "a rapid method to tag a large fraction of the yeast genes" (line 305) then I would think that devising a true screen would be relatively trivial."

This reviewer points also out that an actual screen would be needed to clarify the false positive rate, thereby now referring to the general outcome of a screen. We allow us the remark that the outcome of a screen with respect to false positive rates is strongly dependent on the selection principle that is used for a particular screen, and hence not suitable for a general conclusion with respect to the main aim of this study, namely to describe CASTLING and to characterize the quality and composition of the resulting pooled libraries. We think it is out of question, whether libraries constructed as a pool can serve as valuable resource in a biological meaningful experiment. This has already been shown for example in recent studies which introduced pooled CRISPR-assisted library constructions of less technical challenging genotypes (namely SNPs and small indels) in *Saccharomyces cerevisiae*, which we also reference in the manuscript (PMID:29734294, 29632376, 29786095).

Nevertheless, it is essential to demonstrate that the quantification of genotype distributions in a CASTLING library-based screening experiment can be evaluated properly using Anchor-Seq. This is indeed not sufficiently demonstrated in the experimental data shown in current version of Figure 5. To account for this we now performed a more comprehensive analysis of the the Nanopore Anchor-Seq data. Using this data set, we were able to estimate the protein abundance for 435 proteins. Comparison with all published protein-abundance data sets from yeast demonstrated that pooled GFP libraries made by CASTLING can be profiled with a sensitivity that is comparable to any other method currently used for the quantification of protein abundance of large range of proteins.

In our opinion these results are a very convincing proof that CASTLING libraries together with Anchor-Seq are suitable for sensitive screens, and thus they constitute highly valuable resources for functional gene studies using pooled genotype enrichment strategies.

To further discuss this point and to provide an answer to reviewer #1's question about "true and false positive rates" more specifically in the context of this experiment, it is interesting to note that we have indeed observed genes with higher fluorescence intensity than we would have predicted when we compared the data with the most similar experiment in the literature, the arrayed FACS screen of the GFP collection (Newman et al., Nature 2006, PMID: 16699522). These genes can be thought of as "false positives", and their frequency in our "screen" is about 13% (which, in reverse corresponds to a true positive rate of 87%).

However, instead of detecting false positive clones in our screen, this comparison might as well detect false negative in the use reference data set by Newman and colleagues. Indeed, when comparing the data with all other yeast proteomics datasets in literature (greatly benefiting from the efforts of a recent meta study by Ho et al., Cell Syst, PMID: 29361465) we find cumulative evidence that most of the 'false positives' in our library are in fact 'true positives'. We noted that the 13% outliers exhibiting low abundance estimates in studies using the GFP library, exhibit similar high abundance as our study in other studies independent of the GFP library indicating that these outliers might correspond to 'false negatives' in the GFP library. In this respect it is again interesting to note that the GFP library was constructed before NGS methods that allow the validation of the correctness of clones by sequencing.

We hope that this result convinces the reviewer that his/her speculation that "the absence of a legitimate screen is quite conspicuous, and without it, [...] readers will generally take a wait-and-see approach to the method rather than trying it themselves", does not apply and that the results are convincing enough to motivate readers to try CASTLING out whenever they are interested in a question that cannot be addressed using existing arrayed strain collections.

To account for this new data, we reorganized Figure 5 adding two new panels and one new Supplemental Figure (S10). The following paragraphs in results were modified or inserted to describe the newly added data:

To demonstrate that CASTLING libraries can be used in pooled screening studies, we reverted to fluorescence-activated cell sorting (FACS) which permits sorting based on fluorescence intensity and we used Anchor-Seq for the analysis of the sorted cell pools.

Starting with a library containing 2,052 mNeonGreen tagged genes (Fig. 4c), we first sorted cells for which fluorescence could be detected by FACS. Analysis of the resulting fluorescence-enriched cell population by Anchor-Seq revealed that 848 tagged genes were enriched and 732 tagged-genes were depleted in comparison to the starting library. Based on this analysis we estimated that 35% of the GFP tagged genes in a pooled study can be profiled based on fluorescence intensities which agrees well with a meta-study on yeast protein abundance³⁸ that considered the abundance estimates for 1,404 proteins from fluorescence flow cytometric measurements of individual GFP tagged strains by Newman et al.³⁹ corresponding to 34% of the 4,159 proteins tagged in the GFP library³² that was used for this study

To determine the fluorescence intensity of individual tagged proteins in the fluorescence-enriched fraction, we used FACS and sorted the cells into 8 fractions of increasing fluorescence intensity ('bins', Fig. 5d). Next, we analyzed the genotype distribution within the bins using Anchor-Seq. We sequenced the amplified insertion junctions using MinION nanopore sequencing. This method provides a lower sequencing depth as compared to Illumina dye sequencing, which we usually used to characterize CASTLING libraries. We obtained 18,638 informative reads, which enabled us to determine the relative enrichment in the individual bins for 435 (50%) of the 848 tagged proteins. Comparison of our data with the flow cytometry data set from Newman et al.³⁹ revealed a high correlation (Spearman correlation coefficient of 0.63; Fig. 5e). This value is in the range typically observed when comparing protein abundance data sets, as revealed by a meta-study of yeast protein abundance³⁸ (Supplementary Fig. 10). To estimate whether our small-scale showcase experiment can be considered representative for larger-scale CASTLING library-based experiments we investigated how correlation coefficient changes as a function of the size of compared data sets. Using the data cumulated in the yeast proteome meta-study³⁸ we found that the pairwise correlation coefficients for proteins from our group of 435 tagged genes correlated well with the values estimated when using the full data sets (Pearson correlation coefficient of 0.74; Fig. 5f). This confirms the predictive value of our small-scale study.

While conducting these comparisons, we found that 23 (13%) of 175 tagged genes that were also investigated in the GFP study by Newman et al.³⁹ were apparently fluorescent in our study, but not detected in their study (Fig. 5e, orange dots). Many of these 'false positives' were however reported as being 'abundant' in other studies based on complementary detection approaches such as mass spectrometry⁴⁰ or immuno-blotting against the TAP tag⁴, indicating that they could in fact have resulted from false negative clones of the GFP library (Fig. 5e). An alternative explanation is the increased detection sensitivity provided by the superior green fluorescent protein used in our study (for comparison of different green fluorescent proteins, see ref¹²). Using independently generated clones based on a different gene tagging strategy¹², we could validate the expression of most of these genes when tagged with a fluorescent protein. In addition, we validated by individual tagging the expression of 8 genes detected in our

study that were not represented by the C-GFP library and not characterized in any of the other studies analyzed by Ho et al.³⁸ (Fig. 5g, Supplementary Table 3).

Together, this experiment validates a rapid workflow for profiling experiments using pooled CASTLING libraries. It demonstrates that these libraries constitute a valid resource suitable for phenotypic profiling or screening purposes when combined with Anchor-Seq to analyze the clone distribution across subpopulations isolated from such libraries.

In the discussion we added another paragraph where we discuss the new results and CASTLING library applications (in green) in more details:

CASTLING is a method for gene tagging, and the type of screen that can be performed with such libraries entirely depends on the used tag. Therefore, it is up to the creativity of the researcher to develop a screening procedure to convert the information provided by the tags into information about the biological question in mind. Importantly, a screening procedure requires physical fractionation of the library into sub-pools based on a suitable phenotypic read-out, for example using tags that enable the coupling of a protein behavior such as protein localization⁴⁸ or protein-protein interactions¹⁰ with a growth phenotype.

In our opinion, fluorescent protein reporters constitute a particularly attractive group of tags as they provide visual insights in the cellular organization and dynamics, changes of which are associated with many disturbances of biological processes. Our simple FACS enrichment experiment (Fig. 5d–g) can serve but as proof of principle in this regard as current flow cytometry-based cell sorters cannot resolve more complex cellular phenotypes such as the subcellular localization of proteins⁴⁹. Therefore, the recent development of image-activated cell sorting²³ can be considered a breakthrough to study e.g. how the localization of thousands of proteins is changed under different conditions. We think that for such methods, CASTLING constitutes an important development enabling a variety of entirely new experimental designs and analyses, ranging from functional genomics to biomedical research, paving the way to a new paradigm of shot-gun cell biology.

Finally, the reviewer now also requests to speculate about applications of CASTLING in mammalian cell systems:

“Less important, but I reiterate that I think the authors are doing themselves a disservice by not speculating (which is fine to do in a discussion!) on how those who work in mammalian cell systems could think about applying this approach.”

Answer:

We are not aware that this reviewer has earlier expressed the request to discuss the application of CASTLING in mammalian cells. However, to account for this request we now include a corresponding statement to account for the possibility that CASTLING does work in mammalian cells.

Beyond yeast, CASTLING could be adapted for other organisms able to repair DNA lesions by homologous recombination, including bacteria, fungi, flies and worms, and potentially also in plants and mammalian cells. First evidence that this is the case is provided in Fueller et al. (pre-print, <https://doi.org/10.1101/473876>) where we show that an adapted SIC strategy can be used for efficient endogenous tagging of genes in mammalian cells. We have preliminary data suggesting that CASTLING also works in mammalian cells, although the size of the library that can be generated with it is presently unclear.

In summary, our work shows that CASTLING libraries and quantitative genotype analysis using AnchorSeq seamlessly integrate into existing (and upcoming) high-throughput cell sorting instrumentation to enable functional analyses of pooled resources. This outlines new avenues for the investigation of complex cellular processes in direct competition with strategies based on arrayed library resources.

Reviewer #2 (Remarks to the Author):

In the revised manuscript by Buchmiller et al., the authors addressed all of my specific points. I have no further concerns and fully support publication of this manuscript.

We thank the reviewer for her or his approval.

Additional changes in the manuscript

- We noticed an error in the title of a plot in Fig. 3d. We changed the title of Figure 3d from “oligonucleotide copy number after recombineering” to “[...] after PCR” to stress the intent.
- When preparing the source_data file we noticed missing information for Figure 3f. For this we added the section *Calculation of copy number changes during RCA* on the normalization procedure used for Figure 3f to the Materials & Methods section.
- We corrected a typo in Figure 3g: “library 1b” to “library 2b”.
- We reformatted Supplementary Table 2 to match the labeling of the different libraries in Figure 5 (LibA, LibB, LibC)
- We expanded the methods section to include a description of the analysis of the Nanopore sequencing data for the new Figure panels in Figure 5,e+f
- We optimized the wording and grammar at a few other instances throughout the manuscript.

Reviewers' Comments:

Reviewer #1:

Remarks to the Author:

I have no further comments.